# OPERATOR DEEP SMOOTHING FOR IMPLIED VOLATILITY

**Ruben Wiedemann**
Imperial College London
r.wiedemann22@imperial.ac.uk

**Antoine Jacquier**
Imperial College London
a.jacquier@imperial.ac.uk

**Lukas Gonon**
University of St. Gallen
lukas.gonon@unisg.ch

## ABSTRACT

We devise a novel method for nowcasting implied volatility based on neural operators. Better known as implied volatility smoothing in the financial industry, nowcasting of implied volatility means constructing a smooth surface that is consistent with the prices presently observed on a given option market. Option price data arises highly dynamically in ever-changing spatial configurations, which poses a major limitation to foundational machine learning approaches using classical neural networks. While large models in language and image processing deliver breakthrough results on vast corpora of raw data, in financial engineering the generalization from big historical datasets has been hindered by the need for considerable data pre-processing. In particular, implied volatility smoothing has remained an instance-by-instance, hands-on process both for neural network-based and traditional parametric strategies. Our general *operator deep smoothing* approach, instead, directly maps observed data to smoothed surfaces. We adapt the graph neural operator architecture to do so with high accuracy on ten years of raw intraday S&P 500 options data, using a single model instance. The trained operator adheres to critical no-arbitrage constraints and is robust with respect to subsampling of inputs (occurring in practice in the context of outlier removal). We provide extensive historical benchmarks and showcase the generalization capability of our approach in a comparison with classical neural networks and SVI, an industry standard parametrization for implied volatility. The operator deep smoothing approach thus opens up the use of neural networks on large historical datasets in financial engineering.

## 1 INTRODUCTION

Options trading experienced phenomenal growth in recent years. In its 2023 trading volume report (Cboe Global Markets, Inc., 2024), the CBOE announced the fourth consecutive year of record-breaking volumes on its options exchanges, citing an all-time high number of transactions for European options on the S&P 500 index. European options are financial derivative contracts that give their holder the right, but not the obligation, to either buy or sell an underlying asset at a predetermined price (the *strike*) at a predetermined time (the *expiry*). An option specifying the right to buy (respectively to sell) is called a *Call* (respectively *Put*) option. Options are traded on a wide range of underlyings, including stocks, indices, currencies and commodities, and can be used to hedge against or speculate on the price movements of the underlying asset.

A key concept in options trading is the so-called *implied volatility*, which transforms the nominal price of an option into a conceptually and numerically convenient metric. The *implied volatility surface* is the collection of implied volatilities as observed at a specific point in time, visualized as a surface over the (strike, expiry)-domain. It provides an intuitive representation of the current state of the options market and is crucial for hedging and risk management. The extraction of a smooth surface from quoted option prices is called *implied volatility smoothing* and allows to infer

(or *nowcast*) theoretical option prices for interpolated strike values and expiry times. It remains one of the key challenges in options trading.

Conventionally, implied volatility smoothing relies on parametric surfaces whose parameters are optimized based on the distance to observed prices while adhering to *absence-of-arbitrage* conditions, which ensure the consistency of prices extrapolated from the smoothed surface. The development of such ad-hoc models for implied volatility traces back to *SVI* (Gatheral, 2004), which models implied volatility slice-wise for each expiry and successfully captures its key features on Equity indices. A continuous interpolation scheme for SVI slices was provided in Gatheral & Jacquier (2014), yielding a full surface. Nowadays, sophisticated market makers employ custom parametrizations, which can be considered proprietary trading secrets and reduce SVI to a benchmark role.

Regardless of the particular parametric surface model used, the conventional smoothing approach boils down to the continued execution of a numerical optimization routine: A smoothed surface expires as soon as quotes are updated (whenever markets move), necessitating the re-calibration of parameters. Success and duration of this routine is sensitive to initial conditions, search heuristics, and termination criteria, which exposes practitioners to considerable process uncertainties during trading hours (or *online*). In response, we introduce a novel *operator deep smoothing* approach, replacing the instance-by-instance optimization with a single evaluation of a neural network. This greatly simplifies online calibration, at the upfront cost of training the network *offline* from historical data (in the spirit of Hernandez (2016); Horvath et al. (2021); Liu et al. (2019)). Our unique use of *neural operators* (Kovachki et al., 2023) is fundamentally directed by the observation that the raw inputs for volatility smoothing (the collections of observed volatilities) over time vary in size and spatial arrangement: Options expire, new expiries and strikes become available (the total number of options listed is steadily increasing, see Figure 10), and the coordinates of existing options evolve continuously in the domain of the implied volatility surface (illustrated in Figure 1b and Figure 9). This setting excludes classical neural networks – which required fixed-size inputs – from direct application. Neural operators, instead, conceptualize observed data as point-wise discretizations of latent functions in implicit infinite-dimensional function spaces and are well suited for the task.

**Contributions** We introduce *operator deep smoothing*, a general approach for discretization-invariant data interpolation based on neural operators, and apply it to implied volatility smoothing. Our technique transcends traditional parametric smoothing and directly maps observed volatilities to smoothed surfaces. Comparable neural network based approaches are limited to certain option markets (e.g. FX markets, where options by default spread out on fixed rectilinear grids, as relied upon by Bergeron et al. (2021) for its VAE approach) or require data pre-processing (as in Cont & Vuletić (2023), which achieves fixed rectilinear grids by linear interpolation of market values, setting aside questions related to no-arbitrage constraints). Instead, our technique novelly adapts the *graph neural operator* (GNO) architecture (Anandkumar et al., 2020) to consistently smooth input data of any size and spatial arrangement. While neural operators have successfully been used in Physics to numerically solve partial differential equations, our application is the first in financial engineering and highlights the values of their *discretization-invariance* properties, so far rather under-explored. We employ our method on ten years of intraday S&P 500 options data, smoothing more than 60 million volatility datapoints using a single model instance with around 100 thousand trained parameters. We report errors that substantially improve on the SVI industry benchmark and are highly competitive with Ackerer et al. (2020), which performs smoothing by training one classical neural network per volatility surface. We proceed to successfully demonstrate the generalization capabilities of our model for end-of-day options data of the S&P 500 as well as three further major US indices. No data from these three indices has been used for training.

We explore the technical implications and limitations of our method in Section 5. Here we discuss the broader impact of our contributions:

• Operator Deep Smoothing for Implied Volatility – Our method massively simplifies volatility smoothing, an area where effective methods mean competitive advantage and frequently remain trade secrets. We believe that our approach lowers the entry barrier for effective volatility smoothing, even among industry professionals. Practitioners and researchers who are not directly involved in options trading frequently employ rudimentary methods (SVI or linear/spline interpolation). Here, our trained operator, served as a hands-off tool, could provide "cheap" and accurate surfaces for use in downstream tasks. Ultimately, our method may be useful for all participants of option mar-

kets. This includes the general public, whose trading in such markets has been increasing substantially (Doherty et al., 2023) and which benefits from broadly accessible investment tools.

• Neural Operators for Discretization-Invariant Interpolation – Our operator deep smoothing approach constitutes the first application of neural operators for interpolation/extrapolation tasks and paves the way to future research on the versatility of the discretization-invariance of neural operators in industrial applications characterized by dynamic and spatially irregular data. At least in financial engineering, surfaces analogous to the implied volatility surface (e.g. higher-dimensional equivalents, as, for example, the *volatility cube*) are ubiquitous. We expect our technique to be transferable and to streamline and robustify engineers' and researchers' algorithms and data pipelines.

**Literature review**   The aforementioned SVI was developed for internal use at Merrill Lynch in 1999 and later advocated in Gatheral (2004). Its extension to surface-based SSVI in Gatheral & Jacquier (2014) has been eagerly adopted by practitioners, which have since contributed to its robust calibration and generalizations (Corbetta et al., 2019; Hendriks & Martini, 2017; Guo et al., 2016). It was augmented in Ackerer et al. (2020) by a multiplicative neural network corrector, based on guided network training by means of no-arbitrage soft constraints from Zheng (2018). The absence-of-arbitrage conditions for implied volatility surfaces – providing safeguards for option pricing – were formulated in Roper (2010), and we provide an equivalent formulation, based on Fukasawa (2012); Lucic (2021), for practical purposes. In Chataigner et al. (2020) static arbitrage constraints were used to perform option calibration (with an additional regularization technique), which can be considered to be instance-by-instance smoothing of nominal price data. In Bergeron et al. (2021) a classical VAE (variational autoencoder) was applied to implied volatility smoothing on FX markets, where strikes of quoted options are tied to a fixed grid of *deltas*.[1] This specificity of FX markets allows the use of a conventional feedforward neural network based decoder. Recent option calibration approaches based on neural networks have been proposed in Baschetti et al. (2024); Hernandez (2016); Horvath et al. (2021); Van Mieghem et al. (2023).

A comprehensive account on neural operators is given in Kovachki et al. (2023), unifying previous research on different neural operator architectures and techniques (Anandkumar et al., 2020; Li et al., 2020). Subsequent developments investigating the expressivity of these architectures as well as their generalizations include Hao et al. (2023); Huang et al. (2024); Lanthaler et al. (2023); Li et al. (2021); Lingsch et al. (2023); Tran et al. (2021).

**Outline**   We review financial concepts and the challenges of implied volatility smoothing in Section 2. In Section 3, we provide a review of neural operators (Section 3.1) and introduce our operator deep smoothing approach for general interpolation tasks (Section 3.2). In Section 4, we perform experiments for implied volatility smoothing of S&P 500 options data. Finally, Section 5 gathers technical implications and limitations as well as outlooks regarding the use of neural operators for interpolation purposes.

**Code**   We make code for the paper available at the location `https://github.com/rwicl/operator-deep-smoothing-for-implied-volatility`. In particular, the code repository contains a general *PyTorch* (Paszke et al., 2019) implementation of the graph neural operator architecture for operator deep smoothing.

## 2   BACKGROUND: IMPLIED VOLATILITY

We consider a market of European options written on an underlying asset, which we observe at a given instant $T_0$. We denote the time-$T$ forward price of the underlying asset by $F_{T_0,T}$.

**European Call options**   The market consists of a finite collection of European Call options,[2] each identified by its expiry $T \in (T_0, \infty)$ and its strike $K \in (0, \infty)$. We write $C(T, K)$ for its (undiscounted) price. In practice, these are traded for fixed expiries $T_1, \ldots, T_m$; for each $T_i$, only a finite range of strikes $K_1^i, \ldots, K_{n_i}^i$ is available, widening as the expiry increases (Figure 1).

---

[1]Delta is the derivative of the price with respect to the underlying asset and is standard in FX strike quoting.

[2]In practice, market participants trade both Call and Put options, which are mathematically equivalent through the well-known *Put-Call parity*. The latter thus allows to speak in terms of Call options only.

**Black-Scholes**   The *Black-Scholes model* is the simplest diffusive asset model and captures the volatility of the underlying asset with a single parameter $v \in (0, \infty)$. Its popularity stems from the closed-form expression it admits for the price of a European Call option with *time-to-expiry* $\tau = T - T_0$ and *log-moneyness* $k = \log(K/F_{T_0,T})$:

$$\mathrm{BS}(\tau, k, v) = \Phi\left(d_1\left(\tau, k, v\right)\right) - e^k \, \Phi\left(d_2\left(\tau, k, v\right)\right)$$

(in units of the time-$T$ forward of the underlying). Here, $\Phi$ denotes the cumulative distribution function of the standard Normal distribution, while

$$d_1(\tau, k, v) = \frac{-k}{v\sqrt{\tau}} + \frac{1}{2}v\sqrt{\tau}, \quad d_2(\tau, k, v) = \frac{-k}{v\sqrt{\tau}} - \frac{1}{2}v\sqrt{\tau}. \tag{1}$$

**Implied volatility**   While not able (any longer) to fit market data, the mathematical tractability of the Black-Scholes model gave rise to the concept of *implied volatility*: Given a Call option with price $C(T, K)$, its implied volatility $v(\tau, k)$ is defined by $C(T, K) = F_{T_0,T} \, \mathrm{BS}(\tau, k, v(\tau, k))$. By using time-to-expiry/log-moneyness coordinates, implied volatility provides a universal way to consistently compare the relative expensiveness of options across different strikes, expiries, underlyings, and interest rate environments. Its characteristic shape helps traders make intuitive sense of the states of option markets relative to a (flat) Black-Scholes model baseline.

**Implied volatility smoothing**   This refers to fitting a smooth surface $\hat{v} \colon (0, \infty) \times \mathbb{R} \to (0, \infty)$ to a collection $\mathbf{v} = \{v(\tau_l, k_l)\}_{l=1}^p$ of observed implied volatilities. Naive strategies such as cubic interpolation are ill-fated: Surfaces generated by such interpolation rules will in general correspond to Call option prices that are exploitable by so-called *arbitrage*, namely costless trading strategies generating a guaranteed profit. In option markets, an arbitrage is called *static* if set up solely from fixed positions in options and finitely many rebalancing trades in the underlying. Beyond simple interpolations, practitioners have devised ad-hoc parametrizations for implied volatility (in particular the aforementioned SVI), which are not expected to perfectly match all reference prices. Instead, the model parameters are optimized with respect to an objective function that measures market price discrepancy and includes penalization terms ruling out static arbitrage. These penalization terms are commonly formulated on the basis of the following theorem, which summarizes the shape constraints of the implied volatility surface (Gatheral & Jacquier, 2014; Lucic, 2021; Roper, 2010).

**Theorem 2.1** (Volatility validation). *Let $\hat{v} \colon (0, \infty) \times \mathbb{R} \to (0, \infty)$ be continuous and satisfying:*

(i) *Calendar arbitrage: For each $k \in \mathbb{R}$, $\hat{v}(\cdot, k)\sqrt{\cdot}$ is non-decreasing and vanishes at the origin.*

(ii) *Strike arbitrage: For every $\tau > 0$, the slice $\hat{v}_\tau = \hat{v}(\tau, \cdot)$ is of class $C^2$ with*

$$\mathrm{But}(\tau, \cdot, \hat{v}_\tau, \partial_k \hat{v}_\tau, \partial_k^2 \hat{v}_\tau) \geq 0 \quad \text{and} \quad \limsup_{k \uparrow \infty} \frac{\hat{v}_\tau^2(k)}{k} < \frac{2}{\tau}, \tag{2}$$

*where*

$$\mathrm{But}(\tau, k, v_0, v_1, v_2) = \left(1 + d_1(\tau, k, v_0)v_1\sqrt{\tau}\right)\left(1 + d_2(\tau, k, v_0)v_1\sqrt{\tau}\right) + v_0 v_2 \tau.$$

*Then, $(T, K) \mapsto \mathrm{BS}(\tau, k, \hat{v}(\tau, k))$ defines a Call price surface that is free from static arbitrage.*

Condition 2.1(i) is equivalent to option prices increasing in expiry, while Condition 2.1(ii) arises when computing the *implied probability density* $f_\tau$ of the underlying:

$$f_\tau(\cdot) = \frac{\varphi(-d_2(\tau, \cdot, v))}{v\sqrt{\tau}} \, \mathrm{But}(\tau, \cdot, v, \partial_k v, \partial_k^2 v), \tag{3}$$

where $\varphi$ is the probability density of the standard Normal distribution. Since a probability density needs to be non-negative, equation 3 explains why Condition 2.1(ii) is required.

## 3   Neural operators for discretization-invariant smoothing

### 3.1   Background: neural operators

We provide full details about notations and terms, as well as additional context in Appendix A.

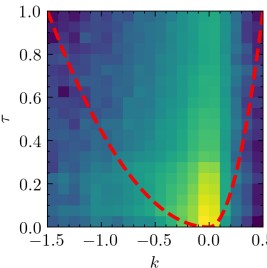 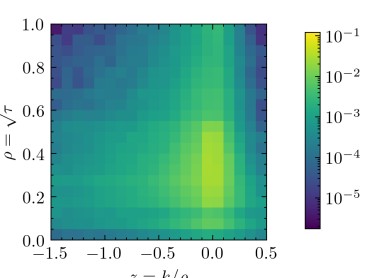 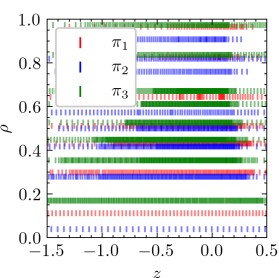

(a) Relative trading volume per quoted interval, averaged over S&P 500 dataset 2012-2021. Left: Rectangular domain w.r.t. time-to-expiry/log-moneyness (99.9% of trading volume with maturities < 1 year). Right: Rectangular domain w.r.t. transformed coordinates (96.6% of trading volume while more evenly populated); boundary delineated in Left.

(b) Scatter plots of example sets of option quotes $\mathbf{v}_1, \mathbf{v}_2, \mathbf{v}_3$. Observed on 15.10.2012, 18.05.2017, and 04.01.2021, respectively, at 10:50:00, each.

Figure 1: Spatial distribution of option quotes in time-to-expiry/log-moneyness domain.

**Philosophy** The development of *neural operators* is based on the philosophy that observed data $\mathbf{a} = \{a_l\}_{l=1}^p$ arises as the evaluation of a latent function $a\colon D \to \mathbb{R}^{c_{\text{in}}}$, defined on some domain $D \subseteq \mathbb{R}^d$, at a discretization $\pi = \{x_l\}_{l=1}^p$ of $D$. That is, $\mathbf{a} = a|_\pi$, or

$$a_l = a(x_l), \quad l = 1, \dots, p. \tag{4}$$

An input-output relationship of data $\mathbf{a} \mapsto \mathbf{u}$ is then "really" described by an operator $F\colon \mathcal{A} \to \mathcal{U}$ between function spaces $\mathcal{A}$ and $\mathcal{U}$. Neural operators are abstract neural network architectures $F^\theta\colon \mathcal{A} \to \mathcal{U}$, with implementations that integrate equation 4 consistently across the variable discretization $\pi$.

**Technicalities** We review the core concepts of neural operators from Kovachki et al. (2023). Let $D$ be a bounded domain in $\mathbb{R}^d$ and $\mathcal{A}$ and $\mathcal{U}$ be Banach spaces of functions mapping from $D$ to $\mathbb{R}^{c_{\text{in}}}$ and $\mathbb{R}^{c_{\text{out}}}$, respectively. Neural operators are finitely parametrized mappings $F^\theta\colon \mathcal{A} \to \mathcal{U}$ with *universality* for continuous target operators and with *discretization-invariant* implementations $\tilde{F}^\theta$. In the space $C(\mathcal{A}, \mathcal{U})$ topologized by uniform convergence on compacts, the architecture $F^\theta$ is called *universal* if $\{F^\theta\}_{\theta \in \Theta}$ is dense in $C(\mathcal{A}, \mathcal{U})$, with $\Theta$ the parameter set. An implementation of $F^\theta$ is an algorithm $\tilde{F}^\theta$ which accepts observed data $\mathbf{a} = a|_\pi$ and outputs a function $u \in \mathcal{U}$ and is such that $\tilde{F}^\theta_\pi(\cdot) = \tilde{F}^\theta(\cdot|_\pi) \in C(\mathcal{A}, \mathcal{U})$. Now, $\tilde{F}^\theta$ is called *discretization-invariant* if $\lim_{n \uparrow \infty} \tilde{F}^\theta_{\pi^{(n)}} = F^\theta$ in $C(\mathcal{A}, \mathcal{U})$, given a *discrete refinement*[3] of $D$.

Let $K$ be a set of input functions, compact in $\mathcal{A}$, and let $\varepsilon > 0$. In combination, universality and discretization-invariance allow to posit the existence of parameters $\theta$, such that for all $a \in K$,

$$\|\tilde{F}^\theta(a|_\pi) - F(a)\|_\mathcal{U} \le \varepsilon, \tag{5}$$

irrespective of the particular discretization $\pi$ given that the data $\mathbf{a} = a|_\pi$ is scattered sufficiently densely across $D$. The training of neural operators is analogous to the classical finite-dimensional setting. It happens in the context of an implicit *training distribution* $\mu$ on the input space $\mathcal{A}$ and aims at minimization of the *generalization error*

$$R_\mu\colon \theta \mapsto \mathbb{E}_{a \sim \mu} \|\tilde{F}^\theta(a|_\pi) - F(a)\|_\mathcal{U}, \tag{6}$$

through the use of gradient descent methods applied to empirical estimates of equation 6. These estimates are constructed from a *training dataset* $\mathcal{D} = \{(\mathbf{a}^{(i)}, \mathbf{u}^{(i)})\}_{i=1}^n$ of *features* $\mathbf{a}^{(i)} = a^{(i)}|_{\pi^{(i)}}$ and *labels* $\mathbf{u}^{(i)} = F(a^{(i)})|_{\pi^{(i)}}$ on the basis of *(mini) batching* heuristics and are frequently transformed or augmented by additional terms through the use of apposite *loss functions*.

### 3.2 OPERATOR DEEP SMOOTHING

Let $\mathbf{v} = \{v(x_l)\}_{l=1}^p$ be the collection of observed data, for example implied volatilities as in Section 2. This notation silently adopts the neural operator philosophy, connecting the data point $v(x_l)$

---

[3]A *discrete refinement* of $D$ is a nested sequence $(\pi^{(n)})_{n \in \mathbb{N}}$ of discretizations of $D$ for which for every $\varepsilon > 0$, there exists $N \in \mathbb{N}$ such that $\{\mathbb{B}_{\mathbb{R}^d}(x, \varepsilon)\colon x \in \pi^{(N)}\}$ covers $D$.

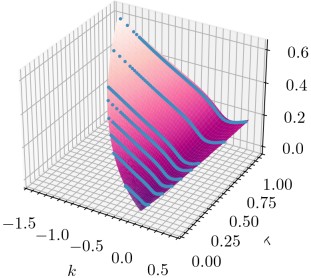 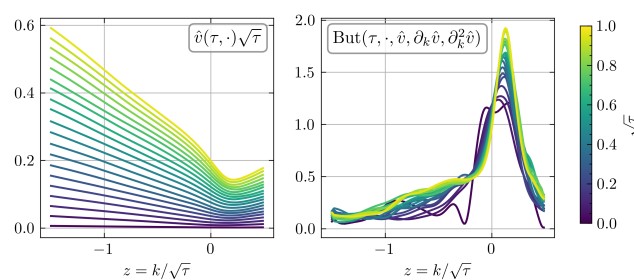

(a) Smoothed surface with input volatility quotes (blue). $\delta_{abs} = 0.005$, $\delta_{spr}(\hat{v}, \mathbf{v}) = 1.020$.

(b) Left: Slices of implied volatility scaled by $\sqrt{\tau}$; absence of crossings indicates absence of calendar arbitrage. Right: Implied probability density factor; positivity indicates absence of butterfly arbitrage. In fact, $\mathcal{L}_{cal}(\theta, \mathbf{v}) = 0$ and $\mathcal{L}_{but}(\theta, \mathbf{v}) = 0$.

Figure 2: Operator deep smoothing of quotes $\mathbf{v}$ from 08.01.2021 at 11:50:00. Compare Figure 13.

with coordinates $x_l$ and hinting at a latent function $v\colon D \to \mathbb{R}$ giving rise to the observed values. The *smoothing* or *interpolation* task consists of constructing an appropriately regular estimate $\hat{v}$ of $v$ from the data $\mathbf{v} = v|_\pi$, with $\pi = \{x_1, \ldots, x_p\}$. The operator deep smoothing approach uses a neural operator $\tilde{F}^\theta$, trained using historical data, to generate $\hat{v} := \tilde{F}^\theta(\mathbf{v})$. It fundamentally leverages the discretization-invariance to produce consistent results even as the "sensors" $x_l$ of the latent function $v$ change in availability and/or location in the domain $D$. This is the situation for volatility smoothing, where the sensors $x_l = (\tau_l, k_l)$ are the (time-to-expiry, log-moneyness) coordinates of the quoted options, which – as noted in Section 1 and illustrated in Figure 1b – move with the market.

**Methodology**  Translating the smoothing task to an operator learning problem, we find that the target operator is the (continuous) identity operator

$$F_{id}\colon C(\overline{D}) \to C(\overline{D}), \quad v \mapsto v,$$

meaning that $F = F_{id}$ and $\mathcal{A} = \mathcal{U} = C(\overline{D})$. The training dataset is the collection $\mathcal{D} = \{\mathbf{v}^{(i)}\}$ of historical data (labels and features coincide), and we train a suitable neural operator architecture $\tilde{F}^\theta$ such that

$$|\tilde{F}^\theta(\mathbf{v})(x) - v(x)| \leq \varepsilon, \quad \text{for all } x \in D, \tag{7}$$

where $v$ denotes the postulated latent function of which we observe $\mathbf{v} = v|_\pi$, and $\varepsilon$ is some given error tolerance for the smoothing task. Instead of minimizing empirical estimates of the $\infty$-norm $\|\tilde{F}^\theta(\mathbf{v}) - v\|_{\mathcal{U}}$, however, we suggest a *fitting loss* based on the *root mean square relative error*,

$$\mathcal{L}_{fit}(\theta, \mathbf{v}) := \sqrt{\frac{1}{|\pi|} \sum_{x \in \pi} \left(\frac{|\tilde{F}^\theta(\mathbf{v})(x) - v(x)|}{1 \vee |v(x)|}\right)^2}, \tag{8}$$

for its smoothness and invariance to the scale of the data.[4] Additional engineering techniques, such as subsampling of inputs during training, are explored in our practical investigation in Section 4.

**Constraints**  Depending on the application, the smoothing task may be subject to constraints. For volatility smoothing, the smoothed surface $\hat{v}^\theta = \tilde{F}^\theta(\mathbf{v})$ must be free of static arbitrage (Theorem 2.1). This is effectively enforced by augmenting the loss function with additional penalization terms, moving away from a pure operator learning problem. This does not only promote the relevant properties in the neural operator output but can also help define it when faced with sparsity of data in the domain $D$ (in this context see also Li et al. (2021)). We handle the strike arbitrage constraint 2.1(ii) via

$$\mathcal{L}_{but}(\theta; \mathbf{v}) = \left\|\left(\text{But}(\cdot, \hat{v}^\theta, \partial_k \hat{v}^\theta, \partial_k^2 \hat{v}^\theta) - \varepsilon\right)^-\right\|_1, \tag{9}$$

---

[4] Empirical estimates of $\infty$-norms and $L^2$-norms are equivalent loss functions on finite-dimensional spaces.

where we ignore the asymptotic condition since our experiments are focused on the bounded domain $D$ (Figure 1a). The inclusion of $\varepsilon$ promotes strictly positive implied densities (we will use $\varepsilon = 10^{-3}$), while we choose the 1-norm to induce sparsity in the constraint violation. The calendar arbitrage constraint 2.1(i) can be tackled analogously with

$$\mathcal{L}_{\text{cal}}(\theta; \mathbf{v}) = \left\| \left( \partial_\tau \left[ (\tau, k) \mapsto v^\theta(\tau, k) \sqrt{\tau} \right] - \varepsilon \right)^- \right\|_1, \tag{10}$$

where again we ignore the asymptotic condition since $D$ is bounded away from zero time-to-expiry.

**Interpolating graph neural operator** Various neural operator architectures exist, mostly arising from the *kernel integral transform* framework of Kovachki et al. (2023). Most prominently, these include Fourier neural operators (FNO), delivering state-of-the-art results on fixed grid data, as well as graph neural operators (GNO), able to handle arbitrary mesh geometries (both reviewed in Appendix A.1). While highly effective with documented universality, these neural operators are not directly applicable for interpolation tasks as their layers include a pointwise-applied linear transformation, which limits the output to the set of the input data locations. Dropping this *local* transformation results in an architecture proved to retain universality Kovachki et al. (2023) and that – at least for its implementation as a GNO – allows to interpolate functions. On the other hand, Lanthaler et al. (2023) proves universality for the architecture combining the local linear transformation with a simple averaging operation, suggesting the fundamental importance of the collaboration of local and non-local components for the expressivity of neural operators. This was noted in Kovachki et al. (2023), for whom retaining the local components can be "beneficial in practice", and confirmed in our experiments, where a purely non-local architecture led to substantially reduced performance.

We therefore propose a new architecture for operator deep smoothing leveraging GNOs' unique ability to handle irregular mesh geometries. We use a purely non-local first layer (dropping the pointwise linear transformation), and use it to produce hidden state at all required output locations, enabling subsequent layers to retain their local transformations. Since GNOs do not theoretically guarantee a smooth output, we augment the training with additional regularization terms such as $\mathcal{L}_{\text{reg}}(\theta; \mathbf{v}) = \|\Delta \hat{v}^\theta\|_2$, with $\Delta$ the Laplace operator, and provide a full description in Appendix B.

## 4 EXPERIMENTS

We detail our practical investigation of the operator deep smoothing approach for implied volatility.

### 4.1 MODEL TRAINING

**Dataset and splits** We perform our numerical experiments using 20-minute interval *CBOE S&P 500 Index Option* data from 2012 to 2021. The dataset amounts to a collection of 49089 implied volatility surfaces and just above 60 million individual volatility datapoints (after domain truncation). We refer the reader to Appendix C.1 for details on the preparation of the dataset. We allocate the first nine years of data (2012 to 2020) to training, keeping 750 randomly drawn surfaces for validation purposes, and use the final year of the dataset (2021) for testing. This yields a training dataset $\mathcal{D}_{\text{train}}$ containing $n_{\text{train}} = 43442$ surfaces, a validation dataset $\mathcal{D}_{\text{val}}$ containing $n_{\text{val}} = 750$ surfaces, and a test dataset $\mathcal{D}_{\text{test}}$ with $n_{\text{test}} = 4897$ surfaces.

**Data transformation** Motivated by Figure 1a, we transform time-to-expiry and log-moneyness via $\rho = \sqrt{\tau}$ and $z = k/\rho$. Intuitively, this transformation converts the "natural" scaling of implied volatility by the square root of time-to-expiry to a scaling of the input domain. From here on, we consider the domain in these coordinates, setting $D = (\rho_{\min}, \rho_{\max}) \times (z_{\min}, z_{\max}) = (0.01, 1) \times (-1.5, 0.5)$. In $(\tau, k)$-coordinates, $D$ becomes a cone-shaped region, that, on average, contains 96.6% of traded options (with time-to-expiry below one year) and, with respect to $(\rho, z)$-coordinates, is more evenly populated, improving the numerics.

**Model configuration** We employ the interpolating GNO as introduced in Section 3.2 and described in detail in Appendix B. The model hyperparameters (giving rise to 102529 trainable parameters in total) were identified by manual experimentation and are detailed in Appendix C.2. We perform ablations for the connectivity of the graph structure (and thus for the tradeoff between expressivity and computational complexity) in Appendix C.6.

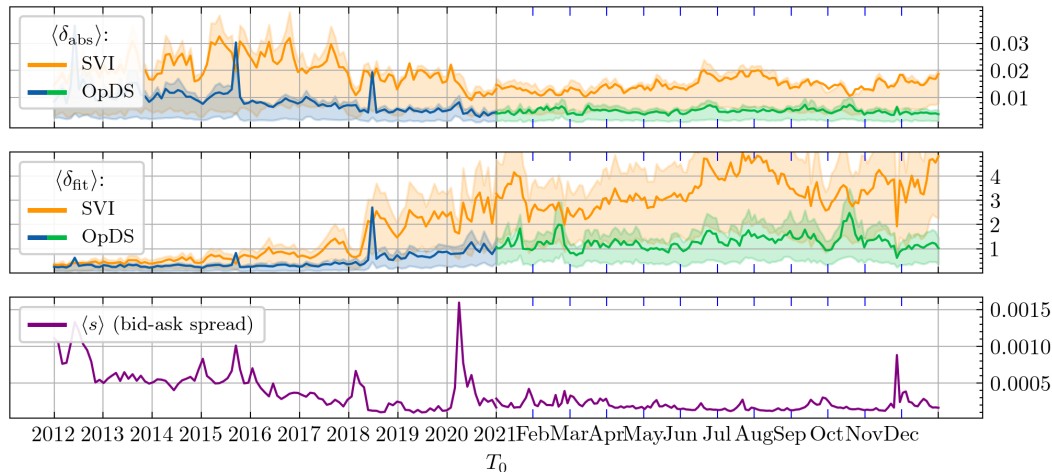

Figure 3: Benchmark of our method (OpDS) vs. SVI, split between training period (left half; 2012 to 2020) and testing period (right half; 2021). Top and middle: Surface-averages $\langle \delta_{\mathrm{abs}} \rangle$ and $\langle \delta_{\mathrm{spr}} \rangle$; shaded regions depict respective interquartile range. Bottom: Surface-average bid-ask spread $\langle s \rangle$.

**Loss function**   We implement a *Vega*-weighted version of $\mathcal{L}_{\mathrm{fit}}$ (equation 8). We compute $\mathcal{L}_{\mathrm{but}}$ (equation 9) using finite differences on a synthetic grid. We use a multiplicative formulation of $\mathcal{L}_{\mathrm{cal}}$ that is invariant to the level of implied volatility. We provide a precise description in Appendix C.3 and perform an ablation study for the weighting of the arbitrage losses in Appendix C.7

**Model training**   We train the GNO for 500 epochs on $\mathcal{D}_{\mathrm{train}}$ using the AdamW optimizer with learning rate $\lambda = 10^{-4}$ and weight decay rate $\beta = 10^{-5}$, and use a pseudo batch size of 64 by accumulating gradients. We randomly subsample the inputs $\mathbf{v}$ and randomize the grids on which we compute the arbitrage losses. The training is performed in around 250 hours using an NVIDIA Quadro RTX 6000 GPU. The validation loss is reported in Table 2.

## 4.2   RESULTS

**Evaluation metrics**   Let $\mathbf{v} = \{v(x)\}_{x \in \pi}$ be a collection of observed implied volatilties and $\hat{v}$ the smoothed surface. We measure absolute relative error:

$$\delta_{\mathrm{abs}}(\hat{v}(x), v(x)) = |\hat{v}(x) - v(x)|/v(x).$$

The average value of $\delta_{\mathrm{abs}}$ over the whole surface $\mathbf{v}$ is the *mean absolute percentage error* (or *MAPE*); we denote it by $\langle \delta_{\mathrm{abs}} \rangle$. As in Corbetta et al. (2019), we moreover realize the importance of analyzing the smoothing algorithm in terms of nominal price error relative to the *bid-ask spread* $s(x) = \mathrm{BS}_{\pm}(x, v_{\mathrm{ask}}(x)) - \mathrm{BS}_{\pm}(x, v_{\mathrm{bid}}(x))$.[5] We define

$$\delta_{\mathrm{spr}}(\hat{v}(x), v(x)) = 2|\mathrm{BS}_{\pm}(x, \hat{v}(x)) - \mathrm{BS}_{\pm}(x, v(x))|/s(x).$$

$\delta_{\mathrm{spr}}(\hat{v}(x), v(x)) \leq 1$ indicates that the prediction $\hat{v}(x)$ for the option $x$ lies within the bid-ask spread.

**Evaluation and model finetuning**   During production use, the GNO would be retrained regularly using the most recent available data. We emulate this procedure for our benchmark in Figure 3. Following the evaluation of the first month of test data (January 2021), the GNO is re-trained for 10 epochs on this data (with each mini-batch augmented by an equal amount of data from the training dataset $\mathcal{D}_{\mathrm{train}}$), before the next month is evaluated. The process is repeated until the entire dataset $\mathcal{D}_{\mathrm{test}}$ is assessed. This finetuning-evaluation procedure takes circa 1.8 GPU hours per month.

---

[5] $v_{\mathrm{bid}}(x)$ and $v_{\mathrm{ask}}(x)$ are the implied volatilities corresponding to Bid and Ask option prices while $\mathrm{BS}_{\pm}$ is the Black-Scholes formula for Call (resp. Put) options for positive (resp. negative) log-moneyness values:

$$\mathrm{BS}_{\pm}(\tau, k, v) = \begin{cases} \Phi(d_1(\tau, k, v)) - e^k \Phi(d_2(\tau, k, v)), & k > 0 \\ e^k \Phi(-d_2(\tau, k, v)) - \Phi(-d_1(\tau, k, v)), & k \leq 0 \end{cases}.$$

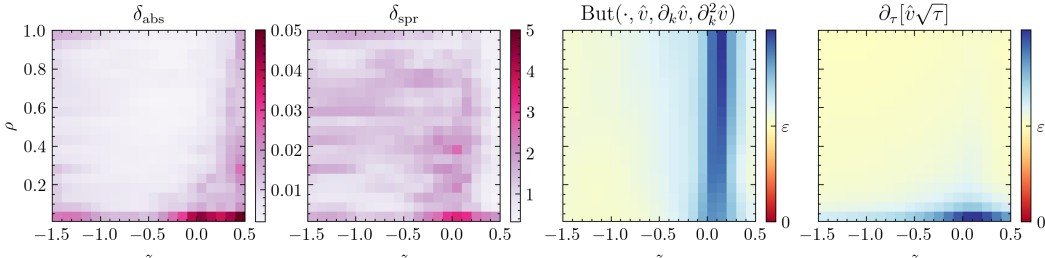

Figure 4: Average spatial distribution of benchmark metrics and arbitrage terms over $\mathcal{D}_{\text{test}}$. The terms $\text{But}(\cdot, \hat{v}, \partial_k \hat{v}, \partial_k^2 \hat{v})$ and $\partial_\tau[\hat{v}\sqrt{\tau}]$ appear in Theorem 2.1. It is apparent that depicted values are strictly positive (indicating absence of arbitrage), lying above the threshold of $\varepsilon = 10^{-3}$.

**Analysis** It is apparent from Figure 3 that our operator approach substantially improves on SVI's smoothing capabilities, with respect to both $\delta_{\text{abs}}$ and $\delta_{\text{spr}}$.[6] Our approach, with monthly finetuning, smooths the volatility surface with a MAPE of around 0.5%, while SVI fluctuates between 1% and 2%. The various figures in Appendix D illustrate the qualitative improvements of our method over SVI. Moreover, our approach appears highly competitive with Ackerer et al. (2020), which performs instance-by-instance volatility smoothing using classical neural networks and reports a MAPE of around 1% for synthetically generated data.[7] We reenact its backtesting for the period Jan-Apr 2018 using our method and summarize the results in Table 5 in Appendix C.5.[8]

An important fact to consider when analyzing Figure 3 is the historical tightening of the bid-ask spread $s$ (displayed in the bottom), driven by increased competition on the S&P 500 option market. This explains why $\delta_{\text{spr}}$ is small early in the training dataset, both for our approach and SVI, while $\delta_{\text{abs}}$ is large: Wide spreads make $\delta_{\text{spr}}$ more lenient an error metric but are accompanied by noisier prices, necessitating greater need for correction by the smoothing algorithm, in turn captured by $\delta_{\text{abs}}$.[9] Similarly, spikes in $s$ (which indicate periods of market stress) help explain spikes in $\delta_{\text{abs}}$.

Complementary to Figure 3, Figure 4 resolves the error metrics as well as the terms controlling the absence of arbitrage spatially, averaged over time. $\delta_{\text{abs}}$ tends to be larger on the Call side (positive log-moneyness), in accordance with Call option's noisier prices (Call options experience less trading than Put options). Moreover, we discern that, on average, the smoothed surfaces are completely free of arbitrage (indicated by non-negativity).

**Generalization** To test the generalization of our approach, we procure end-of-day options data for the S&P 500 (SPX), the NASDAQ-100 (NDX), the Dow Jones Industrial Average (DJX), and the Russell 2000 (RUT) from the *OptionMetrics Ivy DB US* database, accessed by us through the *Wharton Research Data Services* (WRDS). We evaluate the trained operator on the data for the month of January 2021 (right after the training period on the CBOE S&P 500 intraday data), and report the averages of $\langle\delta_{\text{abs}}\rangle$ and $\langle\delta_{\text{spr}}\rangle$ as well as the average arbitrage losses $\mathcal{L}_{\text{cal}}$ and $\mathcal{L}_{\text{but}}$ in Table 1. Firstly, our method maintains its performance on end-of-day S&P 500 data, validating the soundness of our approach: While end-of-day data is slightly different from intraday data, our method still yields small error metrics and arbitrage-free prices. Secondly, the method generalizes well to other indices. We want to stress the fact that our operator has solely been trained on intraday S&P 500 data. Its accurate and virtually arbitrage-free output on end-of-day data of other indices is a strong indicator of the robustness of our approach. We provide further example plots for these datasets in Appendix D.3.

---

[6]We produce the SVI benchmark as described in Section E.

[7]Compare Table 1 of Ackerer et al. (2020), which we note, however, performs smoothing on a larger strike/expiry domain.

[8]We emphasize the following: First, the backtest involves dropping 50% of points for each surface, and our trained operator continues to perform accurate smoothing, a strong indicator of the robustness of our approach with respect to subsampling of inputs. Second, while Ackerer et al. (2020) requires to train 61 neural networks to perform the backtest once, our operator approach enables us to run 25 repetitions in around two minutes on a consumer grade laptop CPU, which is the average time that it takes Ackerer et al. (2020) to train one network.

[9]Compare, e.g., Figure 11 and Figure 13.

Table 1: Average error metrics and arbitrage losses for end-of-day options data for US indices in January 2021. The GNO has been trained solely on intraday S&P 500 data from before 2021.

|  | SPX | NDX | DJX | RUT |
|---|---|---|---|---|
| $\langle \delta_{abs} \rangle$ | 0.00272 | 0.01057 | 0.01629 | 0.00885 |
| $\langle \delta_{spr} \rangle$ | 0.64423 | 1.53306 | 0.20736 | 1.03183 |
| $\mathcal{L}_{but}$ | 6.41e-05 | 2.16e-04 | 1.61e-03 | 4.48e-06 |
| $\mathcal{L}_{cal}$ | 0.00000 | 0.00000 | 3.13e-08 | 0.00000 |

## 5 DISCUSSION

**Summary**   We provide a novel method for implied volatility smoothing, resulting from an application of our general operator approach for discretization-invariant data interpolation. The approach leverages a GNO to directly map given data – consistently across size and spatial arrangement – to smoothed surfaces, transcending classical parametric smoothing techniques. In the example of volatility smoothing, benefits include a massively simplified online calibration process.

**Learning from large datasets**   By moving the application of neural networks from the instance-by-instance level (Ackerer et al., 2020) to the "operator level", we leverage the information contained in the entire training dataset for the smoothing of every single surface. In other words, our method "unlocks" large historical options datasets for volatility smoothing. We argue that our substantial outperformance against Ackerer et al. (2020) in the "Extrapolation-Test"-setting of the benchmark detailed in Table 5 of Appendix C.5 owes to this circumstance.

**Subsampling of inputs**   The discretization-invariance of the GNO entails that our method is robust with respect to subsampling of inputs. In practice, subsampling of inputs occurs in the context of outlier removal. In the example of volatility smoothing, certain quotes may be determined spurious. Simply removing anomalous datapoints from the input is compatible with our method (moreover, we leverage this fact during operator training to improve generalization, compare Appendix C.3).

**Compression**   Figure 3 makes the compression qualities of the operator approach apparent: We compute the entire historical timeseries using a single GNO instance, with around 100 thousand parameters. Evaluating the SVI benchmark, on the other hand, requires 61454 model instances (one per slice), or a total of 307270 parameters. A comparison with Ackerer et al. (2020), which for each smoothed surface trains a new neural network of around 5085 parameters,[10] is striking: Smoothing of the CBOE dataset 2012–2021 at its 20-minute interval frequency would require more than 200 million parameters (more with rising frequency). At the same time, we expect our GNO to perform accurate smoothing over the entire training period and beyond (with regular finetuning), and our model instance remains fixed, even when moving to higher-frequency data.

**Limitations and perspectives**   Compared to ad-hoc volatility parametrizations like SVI, our neural operator approach loses interpretability of parameters, which for some practitioners may be a stringent requirement. This disadvantage is generally shared by neural network based engineering solutions. Moreover, in some situations dimensionality reduction (even without interpretability of parameters) may be a desirable additional feature that is not directly achieved by our operator approach. Combining the VAE method (Bergeron et al., 2021) with our operator approach could lead to further promising potential applications of neural operators. Huang et al. (2024) introduces *neural mappings*, which generalize neural operators to mixed infinite-/finite-dimensionality for input or output spaces. This motivates a discretization-invariant GNO-based encoder, fit to handle raw incoming market data, and a classical decoder to extend the operator approach to a VAE-like architecture.

---

[10]Computed as the sum of $120 = 3 \times 40$ parameters for the input layer, three times $1640 = 41 \times 40$ parameters for the hidden layers, 41 parameters for the output layers, plus 4 additional parameters of the SSVI prior and a scaling parameter.

REPRODUCIBILITY

Primarily, we ensure reproducibility by providing the codebase and model weights used to produce all results in this paper as part of the supplementary material. The codebase includes the data processing components, the GNO architecture, the loss functions, the error metrics, the production of the SVI benchmark, the notebooks used to train and evaluate the models (including hyperparameters and data splits), as well as the notebooks to produce the plots and tables in this paper. The full reproduction of results on intraday data (in particular of Figure 3) is contingent on access to the proprietary CBOE options data, which we are not allowed to provide. In fact, we have stripped the codebase from intermediate benchmarking artifacts that would expose the proprietary data (which some notebooks for the plots rely on). The dataset can be purchased from CBOE, but is expensive. The OptionMetrics end-of-day options data for the suite of indices considered in the final paragraph of Section 4, on the other hand, is more readily and freely available to researchers with subscriptions via the Wharton Research Data Services (WRDS) platform. The provided code allows to directly reproduce the experimental results, in particular, Table 1 and the plots in Appendix D.3. To do so, one would need to download the data from WRDS, persist it at prespecified location detailed in the codebase, and then run the respective notebooks, which automatically load the trained model weights.

To avoid any unclarities in our technique, the Pytorch implementation of our general graph neural operator architecture follows the mathematical definition given in Appendix B as closely as possible. Moreover, the concrete steps undertaken as part of our experiments are detailed in Appendix C:

- Appendix C.1 gives a summary of the processing of the options data.
- The hyperparameter configuration of our model finally employed in our experiments is detailed in Appendix C.2.
- The loss functions and their weights are explicitly defined in Appendix C.3.

ETHICS STATEMENT: DATA AVAILABILITY

The effectiveness of our method is fundamentally tied to the quality and frequency of the options data used for training. Our operator was trained using a proprietary dataset with 20-minute interval frequency, which may not be accessible to many researchers due to its cost. More easily available datasets, such as OptionMetrics data (e.g. freely available from the Wharton Research Data Services), usually only provide daily snapshots of volatility surfaces (end-of-day data). We acknowledge that training using such lower-frequency data may yield an operator that does not have the same smoothing performance as ours. This raises important ethical questions about data accessibility in financial engineering research, where high-quality, high-frequency data is often locked behind paywalls. In our case, the following strategies could potentially mitigate the resulting limitations: Augmenting low-frequency datasets with synthetically generated data based on established parametric models like SVI, or combining data from multiple indices to increase the effective sample size. Practitioners working with lower-frequency data might want to carefully evaluate these approaches and consider the relationship between their data sampling frequency and their intended use case.

ACKNOWLEDGEMENTS

Access to data was made possible through the Clearify Project funding by an Imperial College SME engagement grant jointly with Zeliade Systems, as well as the MSc in Mathematics and Finance, Imperial College London. AJ gratefully acknowledges financial support from the EPSRC grants EP/W032643/1 and EP/T032146/1. RW was supported by the Department of Mathematics, Imperial College, through the Roth scholarship scheme. Computational resources and support were provided by the Imperial College Research Computing Service (Harvey, 2017). For the purpose of open access, the author(s) has applied a Creative Commons Attribution (CC BY) licence (where permitted by UKRI, 'Open Government Licence' or 'Creative Commons Attribution No-derivatives (CC BY-ND) licence' may be stated instead) to any Author Accepted Manuscript version arising.

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

# A    NEURAL OPERATORS

We give a review of the kernel integral transform neural operator framework of Kovachki et al. (2023), and expand in more detail on its graph neural operator.

**Notation and terms**    Let $\mathcal{A}$ and $\mathcal{U}$ be input and output space of an operator learning problem, as introduced in Section 3.1. Then, $\mathcal{A}$ and $\mathcal{U}$ are Banach spaces of functions $D \to \mathbb{R}^{c_{\text{in}}}$ and $D \to \mathbb{R}^{c_{\text{out}}}$, respectively, where $D$ is a (bounded) domain in $\mathbb{R}^d$. The mathematical analysis of neural operators in Kovachki et al. (2023) summarized hereafter, as well as the definition of universality and discretization-invariance in Section 3.1, make use of the following terms and notations.

In the context of Kovachki et al. (2023), a domain is a bounded and connected open set that is topologically regular (in the sense that it is the interior of its closure). A domain $D$ is *Lipschitz* if its boundary locally is the graph of a Lipschitz continuous function defined on an open ball of $\mathbb{R}^{d-1}$. An *open ball* – for any metric space $\mathcal{X} = (\mathcal{X}, d)$ – is the set

$$\mathbb{B}_{\mathcal{X}}(x, \varepsilon) = \{y \in \mathcal{X} : d(y, x) < \varepsilon\}.$$

A *discrete refinement* of $\mathcal{X}$ is a nested sequence $(\pi_n)$ of discretizations of $\mathcal{X}$ (finite subsets of $\mathcal{X}$), such that for every $\varepsilon > 0$ there is $N \in \mathbb{N}$ such that $\{\mathbb{B}(x, \varepsilon) : x \in \pi_N\}$ covers $\mathcal{X}$.

We consider the space $C(\mathcal{A}, \mathcal{U})$ of continuous operators between $\mathcal{A}$ and $\mathcal{U}$. $C(\mathcal{A}, \mathcal{U})$ is topologized by *uniform convergence on compact sets*. With respect to this topology, a sequence $(F_n)_{n \in \mathbb{N}}$ in $C(\mathcal{A}, \mathcal{U})$ converges with limit $F \in C(\mathcal{A}, \mathcal{U})$, if, for every $\varepsilon > 0$ and every compact set $K$ in $\mathcal{A}$, it holds

$$\lim_{n \to \infty} \|F_n - F\|_{\infty, K} = 0.$$

Here,

$$\|H\|_{\infty, K} = \sup_{a \in K} \|H(a)\|_{\mathcal{U}}, \quad H \in C(\mathcal{A}, \mathcal{U}).$$

It is well known that this topology on $C(\mathcal{A}, \mathcal{U})$ is induced by the metric

$$\rho(F, G) = \sum_{n=0}^{\infty} \frac{\|G - F\|_{\infty, \overline{\mathbb{B}}_{C(\mathcal{A}, \mathcal{U})}(0, n)}}{1 \vee \|G - F\|_{\infty, \overline{\mathbb{B}}_{C(\mathcal{A}, \mathcal{U})}(0, n)}}, \quad F, G, \in C(\mathcal{A}, \mathcal{U}).$$

Therefore, the notion of *density* in $C(\mathcal{A}, \mathcal{U})$, as used to define universality of neural operators in Section 3.1, is well defined.

## A.1    KERNEL INTEGRAL NEURAL OPERATORS

**Kernel integral transform neural operators and universality**    A kernel integral transform neural operator consists of the sequential application of:

1) A *lifting layer*

$$L_{\mathcal{P}} \colon [a \colon D \to \mathbb{R}^{c_{\text{in}}}] \mapsto [h_0 \colon D \to \mathbb{R}^{c_0}, \ h_0(x) = \mathcal{P}(a(x))],$$

given by the pointwise application of a function $\mathcal{P} \colon \mathbb{R}^{c_{\text{in}}} \to \mathbb{R}^{c_0}$.

2) The forward propagation through $J$ neural operator layers $L_0, \dots, L_{J-1}$:

$$[h_0 \colon D \to \mathbb{R}^{c_0}] \xmapsto{L_0} [h_1 \colon D \to \mathbb{R}^{c_1}] \xmapsto{L_1} \dots \xmapsto{L_{J-1}} [h_{J-1} \colon D \to \mathbb{R}^{c_J}];$$

each layer $L_j$ operates as

$$h_{j+1}(y) = \sigma_j \left( W_j h_j(y) + \int_D \kappa_j(y, x) h_j(x) dx + b_j(y) \right), \quad y \in D, \tag{11}$$

where

- $W_j \in \mathbb{R}^{c_{j+1} \times c_j}$ is a weight matrix applied pointwise,
- $\kappa_j \in C(D \times D, \mathbb{R}^{c_{j+1} \times c_j})$ is a kernel function parametrizing the integral transform and subject to integrability conditions,
- The bias term $b_j$ is itself a function from $D$ to $\mathbb{R}^{c_{j+1}}$,

- $\sigma_j$ is a classical neural network activation function.

3) A *projection layer*

$$L_{\mathcal{Q}} \colon [h_J \colon D \to \mathbb{R}^{c_J}] \mapsto [u \colon D \to \mathbb{R}^{c_{\text{out}}}, \ u(x) = \mathcal{Q}(h_J(x))],$$

given by the pointwise application of a function $\mathcal{Q} \colon \mathbb{R}^{c_J} \to \mathbb{R}^{c_{\text{out}}}$.

In practice, all components (lifting, kernel functions, projection) are implemented as classical feed-forward neural networks (FFNs). This neural operator architecture is universal in the following sense.

**Theorem A.1** (Universal Approximation; Theorem 11 of Kovachki et al. (2023)). *Let $D$ be a (bounded) Lipschitz domain. Assume:*

- $\mathcal{A} = W^{k_1, p_1}(D)$ *for* $k \in \mathbb{N}_{\geq 0}$ *and* $1 \leq p_1 < \infty$, *or* $\mathcal{A} = C(\overline{D})$.

- $\mathcal{U} = W^{k_2, p_2}(D)$ *for* $k \in \mathbb{N}_{\geq 0}$ *and* $1 \leq p_2 < \infty$, *or* $\mathcal{U} = C(\overline{D})$.

*Then, a subset of kernel integral neural operators, with kernel functions and bias functions taken from a suitable set of FNNs, is dense in $C(\mathcal{A}, \mathcal{U})$.*

**Discretization-invariant implementations** Consider a neural operator $F^\theta$ and let $\pi = \{x_l\}_{l=1}^p$ be a discretization of $D$. To make sense of a basic discretization-invariant implementation for $F^\theta$, associate with $\pi$ a partition $(D_1, \ldots, D_p)$ of $D$ for which $\lambda_d(D_l) > 0$ and $x_l \in D_l$ for $l = 1, \ldots, p$. Here $\lambda_d$ denotes the Lebesgue measure on $\mathbb{R}^d$. Consider the following implementation of $F^\theta$ (written in terms of a single constituent layer $L = (W, \kappa, b, \sigma)$):

$$\tilde{L}(h|_\pi)(y) = \sigma \left( W h(y) + \sum_{l=1}^p \kappa(y, x_l) h(x_l) \lambda_d(D_l) + b(y) \right), \quad y \in D. \tag{12}$$

Kovachki et al. (2023) establishes the following.

**Theorem A.2** (Discretization Invariance; Theorem 8 of Kovachki et al. (2023)). *Let $F^\theta \colon \mathcal{A} \to \mathcal{U}$ be a kernel integral neural operator, where $\mathcal{A}$ and $\mathcal{U}$ both continuously embed into $C(\overline{D})$. Then, the implementation of $F^\theta$ based on equation 12 is discretization-invariant as defined in Section 3.1.*

equation 12 suggests the straightforward (quasi) Monte-Carlo inspired implementation

$$\tilde{L}(h|_\pi)(y) = \sigma \left( W h(y) + \frac{\lambda_d(D)}{|\pi|} \sum_{x \in \pi} \kappa(y, x) h(x) + b(y) \right), \quad y \in D. \tag{13}$$

Most effectively, $\pi$ is a low-discrepancy sequence in $D$.

## A.2 Graph neural operators

The curse of dimensionality makes the direct implementation equation 13 prohibitively expensive in practice. Instead, Anandkumar et al. (2020) introduces *graph neural operators* (or, *GNO*s, for short) which replace the kernel integral operation at the heart of the framework by a sum approximation and organizes the constituent terms using a directed graph structure: The discretization $\pi = \{x_1, \ldots, x_m\}$ of the input data $\mathbf{h} = h|_\pi$ is enriched with a directed graph structure $G_{\mathbf{h}} = (V, E)$, allowing the following implementation $\tilde{F}^\theta$ of $F^\theta$:

$$\tilde{L}(h|_\pi)(y) = \sigma \left( W h(y) + \frac{1}{|\mathcal{N}_{\text{in}}(y)|} \sum_{x \in \mathcal{N}_{\text{in}}(y)} \kappa(y, x) h(x) + b(y) \right), \quad y \in V. \tag{14}$$

Here, $\mathcal{N}_{\text{in}}(y)$ is the set of so-called *in-neighbors* of $y$ in the graph $G_{\mathbf{h}}$: $x \in \mathcal{N}_{\text{in}}(y)$ iff $(x, y) \in E$. It is clear that, to compute output at $y$, the point $y$ must be included as a node into the graph $G_{\mathbf{h}}$. On the other hand, it is necessary to drop the local linear transformation with $W$ if $y$ is not part of the input data locations $\pi$ (compare our discussion in 3). It is important to reconcile the input and output

locations of the layers when creating the graph structure to enable an efficient implementation using *message passing* algorithms.

The graph structure can be meticulously adjusted to implement various complexity-reducing techniques like *Nyström approximation* or *integration domain truncation* that effectively aim at a systematic reduction of the size of $\mathcal{N}_{\text{in}}(y)$; the naive implementation equation 12 is recovered for the case of a complete directed graph (with self-loops) for which $\mathcal{N}_{\text{in}}(y) = \pi$. Note that choosing $\mathcal{N}_{\text{in}}(y)$ as a strict subset of $\pi$ breaks the guaranteed smoothness in the GNO output.

# B    INTERPOLATION GRAPH NEURAL OPERATOR

We detail our modifications of the GNO architecture.

Let $\mathbf{v} = \{v(x)\}_{x \in \pi}$ be the given data.

**Graph construction**    Arguably part of the model architecture is the graph construction: Compile the set $\pi_{\text{out}} = \{y_l\}_{l=0}^q$ of points $y \in D$ at which to compute the smoothed surface $\hat{v}(y)$. During operator training, this will be the set of input locations (to compute the fitting loss) as well as any additional locations needed to compute auxiliary loss terms (the arbitrage losses $\mathcal{L}_{\text{but}}(\theta, \mathbf{v})$) and $\mathcal{L}_{\text{cal}}(\theta, \mathbf{v})$ in the case of volatility smoothing). For each $y \in \pi_{\text{out}}$, we construct the set of in-neighbors from the set of input data locations:

$$\mathcal{N}_{\text{in}}(y) \subseteq \pi_{\text{in}}. \tag{15}$$

In other words, we employ a Nyström approximation with nodes limited to the input data locations. This is an important prerequisite to enable the use of the GNO architecture for interpolation tasks (or, more generally phrased, allows us to employ kernel functions with input skip connections). We set $G_{\mathbf{v}} = (\pi_{\text{out}}, E)$, where

$$E = \bigcup_{y \in \pi_{\text{out}}} \{(x, y) \colon x \in \mathcal{N}_{\text{in}}(y)\}.$$

**Forward propagation**    Given $G_{\mathbf{v}}$, we perform the first step of the forward propagation as follows:

$$\begin{cases} \tilde{h}_0(x) = \mathcal{P}_0(v(x)), & x \in \pi_{\text{in}} \\ h_1(y) = (\sigma_0 \circ \mathcal{Q}_0)\left(\mathcal{K}(\tilde{\mathbf{h}}_0; \mathbf{v})(y) + b_0\right), & y \in \pi_{\text{out}}. \end{cases}$$

For the subsequent layers $j = 1, \dots, J - 1$, we then proceed using the classical scheme:

$$\begin{cases} \tilde{h}_j(y) = \mathcal{P}_j(h_j(y)), \\ h_{j+1}(y) = (\sigma_j \circ \mathcal{Q}_j)\left(W_j \tilde{h}_j(y) + \mathcal{K}_j(\tilde{\mathbf{h}}_j; \mathbf{v})(y) + b_j\right) \end{cases}, \quad y \in \pi_{\text{out}}.$$

In the above:

- $\mathcal{P}_j \colon \mathbb{R}^{c_j} \to \mathbb{R}^{\tilde{c}_j}$ and $\mathcal{Q}_j \colon \mathbb{R}^{\tilde{c}_{j+1}} \to \mathbb{R}^{c_{j+1}}$ are layer-individual lifting and projection, in view of A.1 implemented simply as FNNs.

- $W_j \in \mathbb{R}^{\tilde{c}_{j+1} \times \tilde{c}_j}$ is a weight matrix (not present for $j = 0$), while $b_j \in \mathbb{R}^{\tilde{c}_{j+1}}$ is a constant bias term.

- $\mathcal{K}_j$ is the sum approximation of the kernel integral with kernel weight function $\kappa_j^W \colon D^2 \times \mathbb{R}^{\tilde{c}_j} \times \mathbb{R}^{c_0} \to \mathbb{R}^{\tilde{c}_{j+1} \times \tilde{c}_j}$ and kernel bias function $\kappa_j^b \colon D^2 \times \mathbb{R}^{\tilde{c}_j} \times \mathbb{R}^{c_0} \to \mathbb{R}^{\tilde{c}_{j+1}}$ (both with state and input skip connections):

$$\mathcal{K}_j(\tilde{\mathbf{h}}_j; \mathbf{v})(y) = \frac{1}{|\mathcal{N}_{\text{in}}(y)|} \sum_{x \in \mathcal{N}_{\text{in}}(y)} \kappa_j^W(y, x, \tilde{h}_j(x); v(x))\tilde{h}_j(x) + \kappa_j^b(y, x, \tilde{h}_j(x); v(x)).$$

Both $\kappa_j^W$ and $\kappa_j^b$ are implemented as FNNs in our case, again to satisfy the requirements of A.1 and to keep things simple.

Note that omitting the local linear transformation in the first layer allows to extract the fist hidden state $h_1(y)$ for all $y \in \pi_{\text{out}}$ from the lifted input $\tilde{\mathbf{h}}_0$, which is defined solely for the input locations $x \in \pi_{\text{in}}$. Providing each layer with its own lifting and projection allows to separate the hidden channel size $c_0, \dots, c_J$ from the the dimensions $\tilde{c}_0, \dots, \tilde{c}_J$ of the space in which the integral transformation is performed. Moreover, the individual lifting and projection help re-parametrize the state before performing the integral transform (inspired by the succesful Transformer architecture), which allows to keep the size of the kernel weight matrix low.

## C  SUPPLEMENTARY INFORMATION: DATA, MODEL, TRAINING, EVALUATION

This section contains additional information regarding our empirical study of the operator deep smoothing method for implied volatility smoothing.

### C.1  DATA

**Data source**  Our numerical experiments are based on the "Option Quotes" dataset product available for purchase from the CBOE. Our version of the dataset contains relevant data features for S&P 500 Index options for the years 2012 through 2021 and is summarized on a 20-minute interval basis.

**Data preparation**  We compute *simple mids* for options and underlying by averaging bid and ask quotes and use this aggregate as a reference price for all our subsequent computations. We calculate discount factors and forward prices from Put-Call parity using the industry standard technique based on linear regression. We compute time-to-expiry in units of one year as well as log-moneyness as defined in Section 2. We extract implied volatilities using the *py-vollib-vectorized* project available in the *Python Package Index* at the location `https://pypi.org/project/py-vollib-vectorized/`. *py-vollib-vectorized* implements a vectorized version of Jäckel (2015)'s *Let's-be-rational* state-of-the-art method for computing implied volatility. We discard all implied volatilities of in-the-money options, or, in other words, we compose our implied volatility surface from Put options for non-positive log-moneyness values and Call options for positive log-moneyness values.

### C.2  MODEL

We proceed to detail the hyperparameter configuration of the modified GNO architecture introduced in Appendix B.

**The choice of in-neighborhoods**  The construction of the in-neighborhood sets for the graph neural operator is a crucial hyperparameter choice, fundamentally dictating the computation routes (and thus complexity) of the forward pass of the model. We already explained in Appendix B that we employ a Nyström approximation with subsampling from the input data nodes, to unlock the GNO for interpolation tasks. Additionally, we employ *truncation*. Truncation limits the spatial extent of the in-neighborhoods and is a way to incorporate information about the locality structure of the learning task at hand directly into the graph neural operator architecture. Since implied volatility smoothing requires limited global informational exchange along the time-to-expiry axis, we impose the following restriction on the in-neighborhood sets $\mathcal{N}_{\mathrm{in}}(y)$:

$$\mathcal{N}_{\mathrm{in}}(y) \subseteq \overline{\mathcal{N}_{\mathrm{in}}}(y), \tag{16}$$

where

$$\overline{\mathcal{N}_{\mathrm{in}}}(\rho_y, z_y) = \{(\rho_l, z_l) \in \pi \colon |\rho_l - \rho_y| \leq \overline{\rho}\} \tag{17}$$

is the set of all available options $(\rho_l, z_l)$ contained in the slices with a time-to-expiry $\rho_l$ close than $\overline{\rho}$ to the time-to-expiry $\rho_y$ of $y$. We explain our reasoning more precisely:

- The input data for volatility smoothing is not arbitrarily scattered over the $(\rho, z)$-domain, but arranged as dense $z$-slices that are sparseley distributed along the $\rho$-axis (compare Figure 5 as well as three more examples pictured in Figure 1(b)). Condition 2.1(i) of Theorem 2.1 imposes monotonicity of the output surface along the time-to-expiry axis. This constraint is inherently "local": To generate a compliant output surface, it is sufficient for the hidden states at a given output location to receive information from their immediate neighboring slices.[11] We computed the maximum distance (with respect to $\rho$-coordinates) between slices over our entire dataset as $\Delta_{\max}\rho \approx 0.269$, which is thus established as a lower bound for $\overline{\rho}$, and finally explains our choice $\overline{\rho} = 0.3$. We note that – because we use three hidden GNO layers (see below) – the domain of influence of each input point ultimately is unrestricted: The compositional structure allows information to travel slice to slice in steps of length $\overline{\rho} = 0.3$, which amounts to a total distance of $4 \times 0.3 = 1.2$. This exceeds the size of the considered domain $D = (\rho_{\min}, \rho_{\max}) \times (z_{\min}, z_{\max})$ in the direction of the $\rho$-axis. It is therefore not motivated to increase the level of $\overline{\rho}$.

---

[11] A collection of slices that is monotonously increasing in pairs is montonously increasing as a whole.

- We do not perform a similar truncation in the direction of the log-moneyness axis. In particular, this allows all points in any given slice to connect with each other indiscriminately, which in view of the nonlinear shape constraint 2.1(ii) of Theorem 2.1 is motivated. Moreover, such truncation would limit extrapolation distance in $z$-direction.

To concretely compute $\mathcal{N}_{\text{in}}(y)$ for given $y$ on the basis of equation 16, we employ the following low-discrepancy subsampling heuristic, parametrized by the hyperparameter $K$: First, we compute $\overline{\mathcal{N}_{\text{in}}}(y)$ and convert it to a sequence by sorting it by two-dimensional Euclidean distance to $y$ in ascending order. Of this sequence we take every $k$-th element, where $k$ is the largest step size such that the final number of nodes $\mathcal{N}_{\text{in}}(y)$ does not exceed $K$. This gives us $\mathcal{N}_{\text{in}}(y)$. Note that, by sorting $\overline{\mathcal{N}_{\text{in}}}(y)$ and performing a "sparse" selection, we promote low-discrepancy properties for $\mathcal{N}_{\text{in}}(y)$, which intuitively aid the convergence properties of the kernel integrals (compare 5).

The hyperparameter $K$, finally, constitutes an upper bound on the size of the $\mathcal{N}_{\text{in}}(y)$. It allows us to control the computational complexity of the model, in a trade-off, of course, with the expressivity of the GNO. After manual experimentation, we settle on a value of $K = 50$ and perform an ablation study in Appendix C.6 to validate our choice.

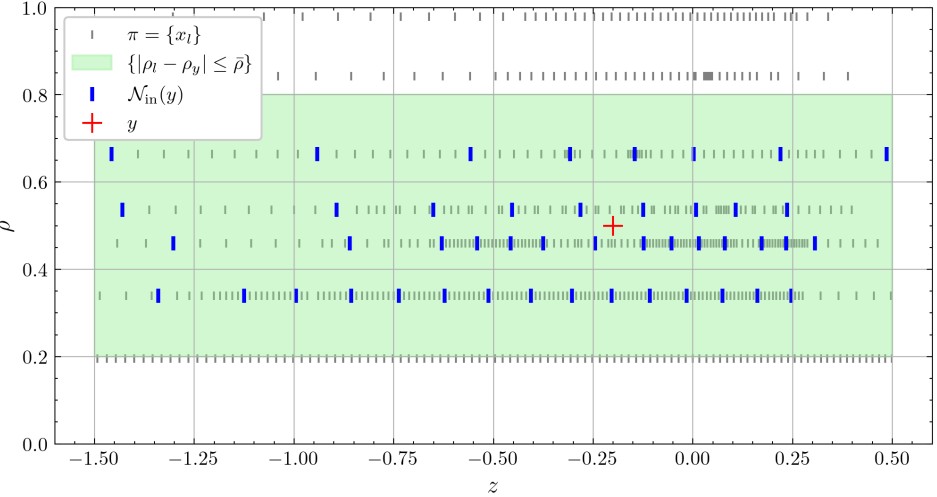

Figure 5: Graph construction: Given data $\mathbf{v} = (v_x)_{x \in \pi}$ (S&P 500 on 10.05.2012 at 16:10:00), generate in-neighborhood for output point $y$ by subsampling from $\overline{\mathcal{N}_{\text{in}}}(\rho_y, z_y)$ by first sorting and then taking a sparse selection, promoting low-discrepancy.

**GNO layers and kernels**  The below choices amount to a total number of 102529 trainable parameters.

- We employ three hidden layers and a channel size of 16: $J = 4$, and $c_1, c_2, c_3 = 16$ ($c_0$ and $c_J$ are determined as 1 by the scalar dimension of volatility data). We use GELU-activations for the hidden layers and a Softplus-activation for the output layer (to ensure the positivity of the smoothed surfaces): $\sigma_0, \ldots, \sigma_{J-1} = \text{GELU}$, and $\sigma_J = \text{Softplus}$.

- We retain $\mathcal{P}_0, \ldots, \mathcal{P}_{J-1}$ and $\mathcal{Q}_J$ as single-hidden layer FNNs with 64 hidden nodes and GELU-activations for the hidden layers. The remaining lifting and projections remain unutilized. In particular, $\tilde{c}_0, \ldots, \tilde{c}_J = 16$.

- We implement the kernel weight and bias functions as two-hidden layer FNNs with 64 hidden nodes and GELU-activations for the hidden layers.

### C.3 TRAINING

**Loss function**  To ease notation we write $\hat{v}^\theta = \tilde{F}^\theta(\mathbf{v})$. We implement a Vega-weighted version of the fitting loss equation 8:

$$\mathcal{L}_{\text{fit}}(\theta; \mathbf{v}) = \left( \frac{1}{|\pi_\mathbf{v}|} \sum_{x \in \pi_\mathbf{v}} w_\mathcal{V}(x; \mathbf{v}) \left| (\hat{v}^\theta(x) - v(x))/v(x) \right|^2 \right)^{1/2}.$$

Here,

$$w_\mathcal{V}(x; \mathbf{v}) = \frac{\mathcal{V}(x, v(x))}{\frac{1}{|\pi|} \sum_{x \in \pi} \mathcal{V}(x, v(x))} \vee 1,$$

where $\mathcal{V}(x, v(x))$ is the *Black-Scholes Vega*, the sensitivity of the Black-Scholes option price with respect to its volatility parameter:

$$\mathcal{V}(\rho, z, v) = \partial_v \text{BS}(\tau, k, v) = \varphi(d_1(\tau, k, v))\sqrt{\tau}. \tag{18}$$

For the implementation of the no-arbitrage penalization terms $\mathcal{L}_{\text{but}}$ and $\mathcal{L}_{\text{cal}}$, we first generate discretizations $\pi_\rho = \{\rho_1, \ldots, \rho_m\}$ and $\pi_z = \{z_1, \ldots, z_n\}$ of $[\rho_{\min}, \rho_{\max}]$ and $[z_{\min}, z_{\max}]$. We resolve the derivative terms $\partial_z v^\theta$ and $\partial_z^2 v^\theta$ on the synthetic rectilinear grid $\pi = \pi_\rho \times \pi_z$ using (central) finite differences. Then, we translate $\mathcal{L}_{\text{but}}$ directly from equation 9 as

$$\mathcal{L}_{\text{but}}(\theta; \mathbf{v}, \pi) = \frac{1}{|\pi|} \sum_{x \in \pi} \left( \text{But}(x; \tilde{v}^\theta(x), \Delta_{z,\pi} \tilde{v}^\theta(x), \Delta_{z,\pi}^2 \tilde{v}^\theta(x)) - \varepsilon \right)^-,$$

where $\text{But}$ is made consistent with the transformed coordinates and we used obvious notation for the finite differences. We use $\varepsilon = 10^{-3}$. On the other hand, we enforce the monotonicity constraint of Theorem 2.1 using

$$\mathcal{L}_{\text{cal}}(\theta; \mathbf{v}, \pi_\rho, \pi_z) = \frac{1}{mn} \sum_{i=1}^m \sum_{j=1}^n \left( \frac{\tilde{v}^\theta(\rho_{i+1}, z_j)}{\tilde{v}^\theta(\rho_i, (\rho_{i+1} z_j)/\rho_i)} - \frac{\rho_i}{\rho_{i+1}} - \varepsilon \right)^-.$$

Compared to a derivative based implementation, this implementation is independent of the scale and – in our empirical experiments – has provided an improved signal. Since the Nyström approximation employed by the graph neural operator (as well as the choice equation 16) break the guaranteed smoothness of the operator output, we additionally introduce $\|\partial_\rho^2 \hat{v}^\theta\|_2$ and $\|\partial_z^2 \hat{v}^\theta\|_2$ as regularization terms:

$$\mathcal{L}_{\text{reg-}\rho}(\theta; \mathbf{v}, \pi) = \sqrt{\frac{1}{|\pi|} \sum_{x \in \pi} |\Delta_{\rho,\pi}^2 \tilde{v}^\theta(x)|^2}, \quad \mathcal{L}_{\text{reg-}z}(\theta; \mathbf{v}, \pi) = \sqrt{\frac{1}{|\pi|} \sum_{x \in \pi} |\Delta_{z,\pi}^2 \tilde{v}^\theta(x)|^2}.$$

We compose the final loss function as a weighted sum of all terms introduced:

$$\mathcal{L}(\theta; \mathbf{v}, \pi_\rho, \pi_z) = \sum \begin{cases} \lambda_{\text{fit}} \mathcal{L}_{\text{fit}}(\theta; \mathbf{v}), \\ \lambda_{\text{but}} \mathcal{L}_{\text{but}}(\theta; \mathbf{v}, \pi_\rho \times \pi_z), \\ \lambda_{\text{cal}} \mathcal{L}_{\text{cal}}(\theta; \mathbf{v}, \pi_\rho, \pi_z), \\ \lambda_{\text{reg-}\rho} \mathcal{L}_{\text{reg-}\rho}(\theta; \mathbf{v}, \pi_\rho \times \pi_z), \\ \lambda_{\text{reg-}z} \mathcal{L}_{\text{reg-}z}(\theta; \mathbf{v}, \pi_\rho \times \pi_z). \end{cases}$$

The specific weights are

| $\lambda_{\text{fit}}$ | $\lambda_{\text{cal}}$ | $\lambda_{\text{but}}$ | $\lambda_{\text{reg-}\rho}$ | $\lambda_{\text{reg-}z}$ |
|---|---|---|---|---|
| 1 | 10 | 10 | 0.01 | 0.01 |

The particular weighting of the individual terms has initially been retrieved by manual experimentation, led by the findings of Ackerer et al. (2020). To additionally validate our choices, we perform an ablation study in Appendix C.7.

**Validation loss**  Table 2 displays descriptive statistics of the validation losses.

Table 2: Validation loss.

|  | mean | std | 1% | 25% | 50% | 75% | 99% |
|---|---|---|---|---|---|---|---|
| $\mathcal{L}$ | 0.0591 | 0.0807 | 0.0351 | 0.0450 | 0.0506 | 0.0578 | 0.1489 |
| $\mathcal{L}_{\text{fit}}$ | 0.0182 | 0.0182 | 0.0066 | 0.0121 | 0.0162 | 0.0203 | 0.0479 |
| $\mathcal{L}_{\text{but}}$ | 0.0000 | 0.0000 | 0.0000 | 0.0000 | 0.0000 | 0.0000 | 0.0000 |
| $\mathcal{L}_{\text{cal}}$ | 0.0001 | 0.0006 | 0.0000 | 0.0000 | 0.0000 | 0.0000 | 0.0001 |
| $\mathcal{L}_{\text{reg-}r}$ | 0.8567 | 2.4065 | 0.3280 | 0.4702 | 0.5788 | 0.7703 | 2.7578 |
| $\mathcal{L}_{\text{reg-}z}$ | 0.7610 | 0.1403 | 0.4815 | 0.6488 | 0.7590 | 0.8679 | 1.0812 |

## C.4 EVALUATION

Here we provide additional results to supplement our performance evaluation. Table 3 and Table 4 display descriptive statistics of our approach (OpDS) versus SVI. OpDS* refers to benchmarking without monthly finetuning. Moreover, we include Figure 6 and Figure 8, which have been created just like Figure 3 and Figure 4 but without monthly finetuning (OpDS*). Finally, Figure 7 shows the average spatial distribution of benchmark metrics and arbitrage term over the training dataset, which complements the same averages on the test dataset shown in Figure 4.

Table 3: Descriptive statistics for surface-MAPE's $\langle \delta_{\text{abs}} \rangle$ over $\mathcal{D}_{\text{val}}/\mathcal{D}_{\text{test}}$.

|  | mean | std | 1% | 25% | 50% | 75% | 99% |
|---|---|---|---|---|---|---|---|
| OpDS | 0.009/0.005 | 0.007/0.001 | 0.003/0.003 | 0.006/0.004 | 0.008/0.005 | 0.010/0.005 | 0.021/0.007 |
| OpDS* | 0.009/0.007 | 0.007/0.001 | 0.003/0.003 | 0.006/0.07 | 0.008/0.007 | 0.010/0.008 | 0.021/0.012 |
| SVI | 0.021/0.015 | 0.006/0.002 | 0.007/0.010 | 0.016/0.013 | 0.020/0.014 | 0.025/0.016 | 0.034/0.020 |

Table 4: Descriptive statistics for surface-averages $\langle \delta_{\text{spr}} \rangle$ over $\mathcal{D}_{\text{val}}/\mathcal{D}_{\text{test}}$.

|  | mean | std | 1% | 25% | 50% | 75% | 99% |
|---|---|---|---|---|---|---|---|
| OpDS | 0.479/1.265 | 0.662/0.347 | 0.193/0.609 | 0.274/1.025 | 0.330/1.240 | 0.550/1.451 | 1.526/2.453 |
| OpDS* | 0.479/1.866 | 0.662/0.574 | 0.193/0.731 | 0.274/1.457 | 0.330/1.826 | 0.550/2.233 | 1.526/3.445 |
| SVI | 1.124/3.382 | 0.877/0.826 | 0.301/1.464 | 0.492/2.827 | 0.715/3.320 | 1.646/3.914 | 3.710/5.247 |

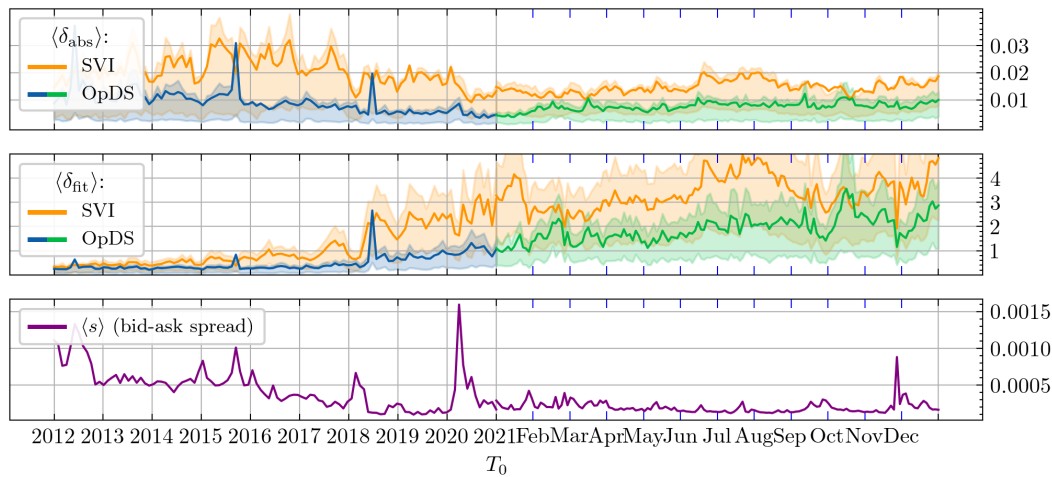

Figure 6: Benchmark comparison between operator deep smoothing without finetuning (OpDS*) and SVI, split between training period (left half; from 2012 to 2020) and testing period (right half; 2021). Surface-averages $\langle \delta_{\text{abs}} \rangle$ (top) and $\langle \delta_{\text{spr}} \rangle$ (middle); shaded region indicate interquartile range. Bottom plot: Surface-average spread $\langle s \rangle$.

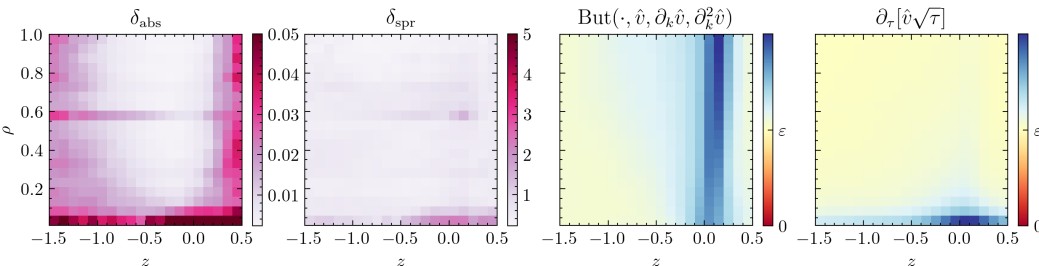

Figure 7: OpDS: Average spatial distribution of benchmark metrics and arbitrage term over train dataset.

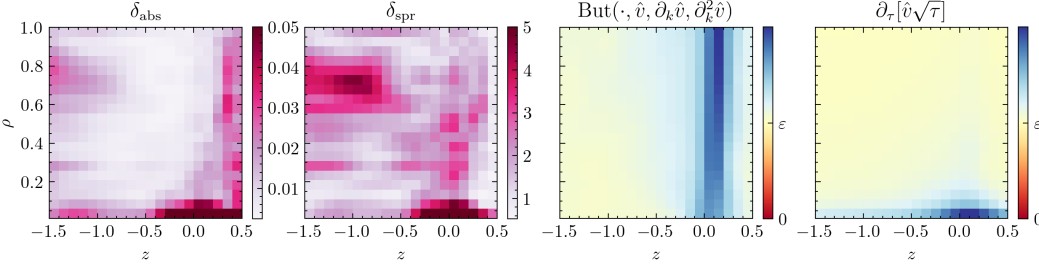

Figure 8: OpDS*: Average spatial distribution of benchmark metrics and arbitrage terms over test dataset (non-finetuned model).

## C.5 COMPARISON TO CLASSICAL NEURAL NETWORKS

We reproduce the experiment underlying Table 1 of Ackerer et al. (2020) using our operator deep smoothing approach. Given observed option quotes, it involves dropping 50% of datapoints, and then measuring the MAPE of the smoothed surface at retained datapoints ("Train") as well as dropped datapoints ("Test"). "Interpolate" and "Extrapolate" are different settings dictating how exactly the datapoints which to drop are selected (for details refer to Ackerer et al. (2020)). The experiment is performed on end-of-day S&P 500 data in the period from January to April 2018 and

averaged percentiles are reported. Table 5 reproduces the relevant row of Table 1 of Ackerer et al. (2020) ("DS") as well as our results averaged over 25 repetitions ("OpDS").

Table 5: Backtesting results ($\langle \delta_{\text{abs}} \rangle$, i.e. MAPE) of Operator Deep Smoothing vs. Deep Smoothing ("DS"; taken from [2]); quantiles in %, Jan-Apr 2018 end-of-day SPX data.

| | | Interpolation | | | | | | Extrapolation | | | | | |
| | | Train | | | Test | | | Train | | | Test | | |
| | $\lambda$ | $q_{05}$ | $q_{50}$ | $q_{95}$ | $q_{05}$ | $q_{50}$ | $q_{95}$ | $q_{05}$ | $q_{50}$ | $q_{95}$ | $q_{05}$ | $q_{50}$ | $q_{95}$ |
|---|---|---|---|---|---|---|---|---|---|---|---|---|---|
| OpDS | 10 | 0.5 | 0.7 | 1.0 | 0.5 | 0.7 | 1.1 | 0.5 | 0.7 | 1.0 | 0.7 | 0.9 | 1.3 |
| DS | 10 | 0.5 | 0.7 | 1.2 | 0.5 | 0.8 | 1.2 | 0.4 | 0.6 | 0.9 | 1.2 | 1.7 | 2.4 |

### C.6 ABLATION: NYSTRÖM APPROXIMATION

We explore the impact of our hyperparameter choice $K = 50$ introduced in Appendix C.2, controlling the size of the Nyström approximation of the integral kernels. We perform an ablation study by resuming training of our trained GNO for the additional values for $K = 3, 5, 10, 20, 30, 40$ as well as $K = 60, 70$. We focus on the data $\mathcal{D}_{2018}$ of the period Jan-Apr 2018, and perform two additional training runs starting from our final GNO-checkpoint (trained for 500 epochs on $\mathcal{D}_{\text{train}}$) as follows:

- 20 epochs each for $K = 40, 30, 20, 10, 5, 3$, in this order. We plan to understand how low $K$ can be for our method to still produce meaningful results.

- 20 epochs each for $K = 60, 70$, in this order. We plan to understand how much additional information the GNO can extract by increasing the value of $K$.

Descriptive statistics for losses and evaluation metrics over $\mathcal{D}_{2018}$ itself are printed in Table 6 and Table 7. We can read from Table 7 that increasing $K$ does not significantly improve performance for either $\delta_{\text{fit}}$ or $\delta_{\text{spr}}$. The mean values for both metrics remain relatively stable for $K > 50$, suggesting diminishing returns with larger $K$ (or, in view of the slightly increasing tendency, the need for additional training). On the other hand, reducing $K$ below its original value of 50 leads to gradual degradation in performance. It is expected that very small $K$-values, especially $K < 10$, result in substantially poorer performance, but it is noteworthy that the progression is quite graceful. Table 6 paints a similar picture for the fitting loss term $\mathcal{L}_{\text{fit}}$, while the auxiliary loss terms slightly increase as $K$ increases. We argue that the increased expressivity awarded by larger values of $K$ leads to slightly more irregular surfaces, and thus to slightly increased arbitrage loss terms. Finally, we argue that our choice of $K = 50$ is validated, where decreasing $K$ is a reasonable strategy when computational resources are scarce and accuracy requirements are not too stringent.

Table 6: Loss terms over $\mathcal{D}_{2018}$ for different values of $K$, after training for 20 additional epochs on $\mathcal{D}_{2018}$.

| | | mean | std | 1% | 25% | 50% | 75% | 99% |
|---|---|---|---|---|---|---|---|---|
| $\mathcal{L}$ | 3 | 0.1113 | 0.0934 | 0.0577 | 0.0819 | 0.0978 | 0.1174 | 0.3771 |
| | 5 | 0.1008 | 0.1022 | 0.0499 | 0.0714 | 0.0849 | 0.1018 | 0.3074 |
| | 10 | 0.0914 | 0.1220 | 0.0412 | 0.0635 | 0.0742 | 0.0885 | 0.2739 |
| | 20 | 0.0854 | 0.1484 | 0.0371 | 0.0580 | 0.0669 | 0.0799 | 0.2802 |
| | 30 | 0.0852 | 0.1834 | 0.0360 | 0.0561 | 0.0650 | 0.0776 | 0.2916 |
| | 40 | 0.0890 | 0.2341 | 0.0367 | 0.0561 | 0.0647 | 0.0779 | 0.3511 |
| | 50 | 0.0906 | 0.2416 | 0.0388 | 0.0564 | 0.0644 | 0.0782 | 0.4900 |
| | 60 | 0.0930 | 0.2956 | 0.0371 | 0.0555 | 0.0644 | 0.0773 | 0.5752 |
| | 70 | 0.0889 | 0.2293 | 0.0371 | 0.0556 | 0.0642 | 0.0773 | 0.4404 |
| $\mathcal{L}_{\text{fit}}$ | 3 | 0.0639 | 0.0346 | 0.0237 | 0.0462 | 0.0575 | 0.0739 | 0.2017 |
| | 5 | 0.0484 | 0.0243 | 0.0195 | 0.0369 | 0.0466 | 0.0557 | 0.1428 |
| | 10 | 0.0354 | 0.0188 | 0.0132 | 0.0283 | 0.0340 | 0.0402 | 0.0681 |
| | 20 | 0.0264 | 0.0168 | 0.0102 | 0.0202 | 0.0245 | 0.0294 | 0.0638 |
| | 30 | 0.0234 | 0.0167 | 0.0088 | 0.0171 | 0.0215 | 0.0258 | 0.0785 |
| | 40 | 0.0220 | 0.0169 | 0.0084 | 0.0157 | 0.0200 | 0.0238 | 0.0963 |
| | 50 | 0.0207 | 0.0174 | 0.0084 | 0.0147 | 0.0183 | 0.0221 | 0.1122 |
| | 60 | 0.0215 | 0.0186 | 0.0083 | 0.0150 | 0.0192 | 0.0230 | 0.1236 |
| | 70 | 0.0217 | 0.0181 | 0.0081 | 0.0152 | 0.0195 | 0.0234 | 0.1218 |
| $\mathcal{L}_{\text{but}}$ | 3 | 0.0000 | 0.0001 | 0.0000 | 0.0000 | 0.0000 | 0.0000 | 0.0001 |
| | 5 | 0.0000 | 0.0005 | 0.0000 | 0.0000 | 0.0000 | 0.0000 | 0.0000 |
| | 10 | 0.0002 | 0.0035 | 0.0000 | 0.0000 | 0.0000 | 0.0000 | 0.0001 |
| | 20 | 0.0003 | 0.0079 | 0.0000 | 0.0000 | 0.0000 | 0.0000 | 0.0000 |
| | 30 | 0.0005 | 0.0117 | 0.0000 | 0.0000 | 0.0000 | 0.0000 | 0.0001 |
| | 40 | 0.0008 | 0.0163 | 0.0000 | 0.0000 | 0.0000 | 0.0000 | 0.0002 |
| | 50 | 0.0009 | 0.0168 | 0.0000 | 0.0000 | 0.0000 | 0.0000 | 0.0010 |
| | 60 | 0.0012 | 0.0216 | 0.0000 | 0.0000 | 0.0000 | 0.0000 | 0.0004 |
| | 70 | 0.0008 | 0.0155 | 0.0000 | 0.0000 | 0.0000 | 0.0000 | 0.0005 |
| $\mathcal{L}_{\text{cal}}$ | 3 | 0.0000 | 0.0006 | 0.0000 | 0.0000 | 0.0000 | 0.0000 | 0.0001 |
| | 5 | 0.0001 | 0.0006 | 0.0000 | 0.0000 | 0.0000 | 0.0000 | 0.0001 |
| | 10 | 0.0001 | 0.0007 | 0.0000 | 0.0000 | 0.0000 | 0.0000 | 0.0000 |
| | 20 | 0.0001 | 0.0007 | 0.0000 | 0.0000 | 0.0000 | 0.0000 | 0.0001 |
| | 30 | 0.0001 | 0.0007 | 0.0000 | 0.0000 | 0.0000 | 0.0000 | 0.0003 |
| | 40 | 0.0001 | 0.0007 | 0.0000 | 0.0000 | 0.0000 | 0.0000 | 0.0004 |
| | 50 | 0.0001 | 0.0008 | 0.0000 | 0.0000 | 0.0000 | 0.0000 | 0.0005 |
| | 60 | 0.0001 | 0.0008 | 0.0000 | 0.0000 | 0.0000 | 0.0000 | 0.0009 |
| | 70 | 0.0001 | 0.0008 | 0.0000 | 0.0000 | 0.0000 | 0.0000 | 0.0009 |
| $\mathcal{L}_{\text{reg-}\rho}$ | 3 | 1.1568 | 2.4882 | 0.3534 | 0.5468 | 0.6905 | 1.0271 | 6.9051 |
| | 5 | 1.3544 | 3.0528 | 0.3823 | 0.5419 | 0.7105 | 1.1629 | 8.1210 |
| | 10 | 1.4194 | 3.3473 | 0.3897 | 0.5763 | 0.8204 | 1.2643 | 8.6570 |
| | 20 | 1.4614 | 3.2880 | 0.3676 | 0.6552 | 0.8875 | 1.3222 | 9.0101 |
| | 30 | 1.5008 | 3.3752 | 0.3790 | 0.7009 | 0.9327 | 1.3734 | 8.8143 |
| | 40 | 1.5716 | 3.5193 | 0.4026 | 0.7334 | 0.9835 | 1.4533 | 9.2937 |
| | 50 | 1.5256 | 3.5376 | 0.4661 | 0.7785 | 1.0336 | 1.4189 | 9.2313 |
| | 60 | 1.5924 | 3.6494 | 0.4175 | 0.7514 | 1.0053 | 1.4776 | 9.4071 |
| | 70 | 1.5610 | 3.4815 | 0.3999 | 0.7508 | 0.9877 | 1.4551 | 8.8669 |
| $\mathcal{L}_{\text{reg-}z}$ | 3 | 0.7183 | 0.0900 | 0.4913 | 0.6589 | 0.7346 | 0.7832 | 0.8723 |
| | 5 | 0.7134 | 0.0788 | 0.5375 | 0.6597 | 0.7193 | 0.7649 | 0.8887 |
| | 10 | 0.7240 | 0.0773 | 0.5572 | 0.6623 | 0.7260 | 0.7773 | 0.8966 |
| | 20 | 0.7413 | 0.1010 | 0.5583 | 0.6585 | 0.7384 | 0.8171 | 0.9738 |
| | 30 | 0.7476 | 0.1177 | 0.5362 | 0.6563 | 0.7420 | 0.8326 | 1.0086 |
| | 40 | 0.7545 | 0.1404 | 0.5280 | 0.6516 | 0.7467 | 0.8425 | 1.0482 |
| | 50 | 0.7616 | 0.1523 | 0.5270 | 0.6533 | 0.7586 | 0.8520 | 1.0745 |
| | 60 | 0.7593 | 0.1647 | 0.5255 | 0.6517 | 0.7542 | 0.8463 | 1.0617 |
| | 70 | 0.7624 | 0.1489 | 0.5286 | 0.6538 | 0.7585 | 0.8552 | 1.0720 |

Table 7: Evaluation metrics over $\mathcal{D}_{2018}$ for different values of $K$, after training for 20 additional epochs on $\mathcal{D}_{2018}$.

|  | $K$ | mean | std | 1% | 25% | 50% | 75% | 99% |
|---|---|---|---|---|---|---|---|---|
| $\langle \delta_{\text{abs}} \rangle$ | 3 | 0.0414 | 0.0232 | 0.0159 | 0.0289 | 0.0376 | 0.0473 | 0.1427 |
|  | 5 | 0.0323 | 0.0140 | 0.0127 | 0.0242 | 0.0310 | 0.0379 | 0.0898 |
|  | 10 | 0.0230 | 0.0093 | 0.0081 | 0.0178 | 0.0221 | 0.0270 | 0.0500 |
|  | 20 | 0.0166 | 0.0078 | 0.0064 | 0.0127 | 0.0156 | 0.0194 | 0.0473 |
|  | 30 | 0.0145 | 0.0079 | 0.0053 | 0.0106 | 0.0135 | 0.0169 | 0.0529 |
|  | 40 | 0.0135 | 0.0083 | 0.0051 | 0.0098 | 0.0126 | 0.0152 | 0.0562 |
|  | 50 | 0.0127 | 0.0087 | 0.0050 | 0.0092 | 0.0116 | 0.0138 | 0.0571 |
|  | 60 | 0.0132 | 0.0101 | 0.0048 | 0.0093 | 0.0120 | 0.0145 | 0.0628 |
|  | 70 | 0.0133 | 0.0095 | 0.0048 | 0.0093 | 0.0121 | 0.0148 | 0.0622 |
| $\langle \delta_{\text{spr}} \rangle$ | 3 | 3.7755 | 4.9387 | 0.8977 | 1.4838 | 2.0117 | 3.2423 | 24.8441 |
|  | 5 | 2.7120 | 3.3901 | 0.6315 | 1.1012 | 1.4551 | 2.4550 | 18.7650 |
|  | 10 | 1.8229 | 2.4049 | 0.4316 | 0.7320 | 0.9282 | 1.7497 | 12.9783 |
|  | 20 | 1.2243 | 1.6054 | 0.3171 | 0.4991 | 0.6227 | 1.1353 | 7.7893 |
|  | 30 | 0.9980 | 1.2846 | 0.2787 | 0.4229 | 0.5180 | 0.9376 | 5.6118 |
|  | 40 | 0.9307 | 1.2340 | 0.2742 | 0.3907 | 0.4768 | 0.8775 | 5.1398 |
|  | 50 | 0.8634 | 1.1477 | 0.2627 | 0.3639 | 0.4356 | 0.8218 | 4.4631 |
|  | 60 | 0.9012 | 1.2789 | 0.2647 | 0.3710 | 0.4483 | 0.8584 | 5.2339 |
|  | 70 | 0.9038 | 1.2559 | 0.2607 | 0.3773 | 0.4570 | 0.8498 | 5.2972 |

## C.7 Ablation: weighting of arbitrage penalties

To assess the impact of weighting of the arbitrage penalties in the loss function, we perform the following experiment: We resume training of our trained GNO for 20 additional epochs on the full training dataset $\mathcal{D}_{\text{train}}$, varying the weights $\lambda_{\text{cal}}$ and $\lambda_{\text{fit}}$ of $\mathcal{L}_{\text{cal}}$ and $\mathcal{L}_{\text{but}}$. More precisely, we equally weight both terms $\mathcal{L}_{\text{cal}}$ and $\mathcal{L}_{\text{but}}$ at the values $\lambda_{\text{arb}} = 0, 1, 10, 100, 1000, 10000$ (we include the original value $\lambda_{\text{arb}} = 10$ to maintain a fair baseline). We start each training run from our final GNO-checkpoint (trained for 500 epochs on $\mathcal{D}_{\text{train}}$ with $\lambda_{\text{arb}} = 10$). The results are reported in Table 8 and Table 9, and we make the following observations:

- The particular choices $\lambda_{\text{arb}}$ affect the achieved loss terms in the expected ways. For a value of $\lambda_{\text{arb}} > 10$ all traces of the arbitrage penalties vanish from the table. At the same time, however, accuracy (as measured by $\mathcal{L}_{\text{fit}}$, $\delta_{\text{abs}}$, and $\delta_{\text{rel}}$) suffers. For choices $\lambda_{\text{arb}} < 10$, it is possible to read a non-zero average for the calendar loss from the table. For $\lambda_{\text{arb}} = 1000$ and $\lambda_{\text{arb}}$. At the same time, however, accuracy (as measured by $\mathcal{L}_{\text{fit}}$, $\delta_{\text{abs}}$, $\delta_{\text{rel}}$) suffers. Our choice $\lambda_{\text{arb}} = 10$ is validated: $\lambda_{\text{arb}} = 1$ or even $\lambda_{\text{arb}} = 0$ do not seem to unlock substantial additional accuracy of the GNO. If there is a small effect, it comes at a cost of increased arbitrage in the smoothed surfaces, as measured by $\mathcal{L}_{\text{cal}}$ and $\mathcal{L}_{\text{but}}$.

- $\lambda_{\text{cal}}$ has a counter-regularizing effect in $\rho$-direction, and we suspect overfitting of the monotonicity constraint. $\mathcal{L}_{\text{but}}$, instead, remains stable for all values. Practitioners will be aware that the calendar arbitrage constraint is usually more demanding than the butterfly arbitrage constraint.

Table 8: Loss terms over $\mathcal{D}_{\text{val}}$ for different values of $\lambda_{\text{arb}}$, after training for 20 additional epochs on $\mathcal{D}_{\text{train}}$.

| | $\lambda_{\text{arb}}$ | mean | std | 1% | 25% | 50% | 75% | 99% |
|---|---|---|---|---|---|---|---|---|
| $\mathcal{L}$ | 0 | 0.0577 | 0.0750 | 0.0341 | 0.0441 | 0.0499 | 0.0567 | 0.1312 |
| | 1 | 0.0575 | 0.0733 | 0.0344 | 0.0441 | 0.0497 | 0.0568 | 0.1375 |
| | 10 | 0.0583 | 0.0774 | 0.0351 | 0.0446 | 0.0501 | 0.0573 | 0.1452 |
| | 100 | 0.0581 | 0.0599 | 0.0352 | 0.0456 | 0.0511 | 0.0580 | 0.1358 |
| | 1000 | 0.0617 | 0.0658 | 0.0380 | 0.0480 | 0.0535 | 0.0609 | 0.1474 |
| | 10000 | 0.1304 | 0.0802 | 0.0825 | 0.1023 | 0.1148 | 0.1381 | 0.2793 |
| $\mathcal{L}_{\text{fit}}$ | 0 | 0.0180 | 0.0181 | 0.0066 | 0.0118 | 0.0160 | 0.0202 | 0.0457 |
| | 1 | 0.0181 | 0.0182 | 0.0063 | 0.0118 | 0.0162 | 0.0202 | 0.0435 |
| | 10 | 0.0182 | 0.0181 | 0.0065 | 0.0120 | 0.0164 | 0.0204 | 0.0469 |
| | 100 | 0.0195 | 0.0315 | 0.0069 | 0.0123 | 0.0165 | 0.0206 | 0.0497 |
| | 1000 | 0.0225 | 0.0479 | 0.0088 | 0.0136 | 0.0177 | 0.0218 | 0.0529 |
| | 10000 | 0.0603 | 0.0669 | 0.0287 | 0.0401 | 0.0474 | 0.0652 | 0.1369 |
| $\mathcal{L}_{\text{but}}$ | 0 | 0.0000 | 0.0000 | 0.0000 | 0.0000 | 0.0000 | 0.0000 | 0.0001 |
| | 1 | 0.0000 | 0.0000 | 0.0000 | 0.0000 | 0.0000 | 0.0000 | 0.0001 |
| | 10 | 0.0000 | 0.0000 | 0.0000 | 0.0000 | 0.0000 | 0.0000 | 0.0000 |
| | 100 | 0.0000 | 0.0000 | 0.0000 | 0.0000 | 0.0000 | 0.0000 | 0.0000 |
| | 1000 | 0.0000 | 0.0000 | 0.0000 | 0.0000 | 0.0000 | 0.0000 | 0.0000 |
| | 10000 | 0.0000 | 0.0000 | 0.0000 | 0.0000 | 0.0000 | 0.0000 | 0.0000 |
| $\mathcal{L}_{\text{cal}}$ | 0 | 0.0001 | 0.0006 | 0.0000 | 0.0000 | 0.0000 | 0.0000 | 0.0001 |
| | 1 | 0.0001 | 0.0006 | 0.0000 | 0.0000 | 0.0000 | 0.0000 | 0.0001 |
| | 10 | 0.0000 | 0.0006 | 0.0000 | 0.0000 | 0.0000 | 0.0000 | 0.0001 |
| | 100 | 0.0000 | 0.0003 | 0.0000 | 0.0000 | 0.0000 | 0.0000 | 0.0000 |
| | 1000 | 0.0000 | 0.0000 | 0.0000 | 0.0000 | 0.0000 | 0.0000 | 0.0000 |
| | 10000 | 0.0000 | 0.0000 | 0.0000 | 0.0000 | 0.0000 | 0.0000 | 0.0000 |
| $\mathcal{L}_{\text{reg-}\rho}$ | 0 | 0.8019 | 2.1876 | 0.3091 | 0.4490 | 0.5489 | 0.7329 | 2.8897 |
| | 1 | 0.7925 | 2.1203 | 0.3102 | 0.4410 | 0.5395 | 0.7268 | 2.9587 |
| | 10 | 0.8236 | 2.2886 | 0.3301 | 0.4578 | 0.5573 | 0.7476 | 2.6687 |
| | 100 | 0.7756 | 1.2831 | 0.3332 | 0.4866 | 0.5858 | 0.7758 | 3.4684 |
| | 1000 | 0.8216 | 0.8114 | 0.3927 | 0.5394 | 0.6450 | 0.8389 | 3.5378 |
| | 10000 | 2.0534 | 0.7765 | 1.2892 | 1.6319 | 1.8667 | 2.2504 | 5.5746 |
| $\mathcal{L}_{\text{reg-}z}$ | 0 | 0.7621 | 0.1394 | 0.4876 | 0.6498 | 0.7599 | 0.8690 | 1.0786 |
| | 1 | 0.7633 | 0.1390 | 0.4876 | 0.6506 | 0.7596 | 0.8695 | 1.0829 |
| | 10 | 0.7572 | 0.1379 | 0.4793 | 0.6477 | 0.7529 | 0.8619 | 1.0702 |
| | 100 | 0.7596 | 0.1409 | 0.4909 | 0.6512 | 0.7554 | 0.8649 | 1.0812 |
| | 1000 | 0.7467 | 0.1339 | 0.4903 | 0.6442 | 0.7414 | 0.8494 | 1.0558 |
| | 10000 | 0.7507 | 0.1462 | 0.5013 | 0.6365 | 0.7422 | 0.8278 | 1.1014 |

Table 9: Evaluation metrics over $\mathcal{D}_{\mathrm{val}}$ for different values of $\lambda_{\mathrm{arb}}$, after training for 20 additional epochs on $\mathcal{D}_{\mathrm{train}}$.

|  |  | mean | std | 1% | 25% | 50% | 75% | 99% |
|---|---|---|---|---|---|---|---|---|
| $\langle\delta_{\mathrm{abs}}\rangle$ | 0 | 0.0108 | 0.0097 | 0.0045 | 0.0073 | 0.0098 | 0.0122 | 0.0265 |
|  | 1 | 0.0109 | 0.0100 | 0.0042 | 0.0071 | 0.0098 | 0.0122 | 0.0262 |
|  | 10 | 0.0110 | 0.0091 | 0.0044 | 0.0073 | 0.0101 | 0.0125 | 0.0260 |
|  | 100 | 0.0121 | 0.0217 | 0.0046 | 0.0074 | 0.0101 | 0.0125 | 0.0276 |
|  | 1000 | 0.0139 | 0.0291 | 0.0057 | 0.0086 | 0.0109 | 0.0134 | 0.0302 |
|  | 10000 | 0.0388 | 0.0360 | 0.0182 | 0.0265 | 0.0315 | 0.0419 | 0.0923 |
| $\langle\delta_{\mathrm{spr}}\rangle$ | 0 | 0.6138 | 1.0777 | 0.2218 | 0.3130 | 0.3728 | 0.7277 | 1.9947 |
|  | 1 | 0.6039 | 1.0790 | 0.2221 | 0.3062 | 0.3685 | 0.7026 | 1.9557 |
|  | 10 | 0.6186 | 1.0552 | 0.2200 | 0.3080 | 0.3775 | 0.7475 | 2.1704 |
|  | 100 | 0.7241 | 2.7257 | 0.2227 | 0.3137 | 0.3797 | 0.7378 | 2.4623 |
|  | 1000 | 0.9303 | 4.1817 | 0.2399 | 0.3480 | 0.4265 | 0.9299 | 2.8245 |
|  | 10000 | 3.1238 | 7.3292 | 0.5128 | 0.8566 | 1.3404 | 3.6653 | 14.2364 |

## C.8 DISCRETIZATION-INVARIANCE IN THE CONTEXT OF IMPLIED VOLATILITY SMOOTHING

Figure 9 illustrates the fact that the coordinates of options continuously evolve with respect to their time-to-expiry/log-moneyness coordinates. It shows the trajectory of three S&P 500 Put options expiring on 19.06.2020, with strike prices of $K = 2000, 3000, 4000$, over the period from 01.06.2019 to 01.06.2020. The 2020 (Covid) stock market crash and subsequent recovery and the sudden increase in log-moneyness of the options is clearly discernible. Moreover, over the lifetime of these options, all three options leave our smoothing domain (one temporarily), which means that from that point onward these options are dropped from the input of the GNO (and one returns). Figure 10, instead, shows that the number of listed options to be processed by the GNO has continuously been increasing. It is the discretization-invariance of the GNO architecture that allows our method to produce consistent results over the whole timeline of Figure 10.

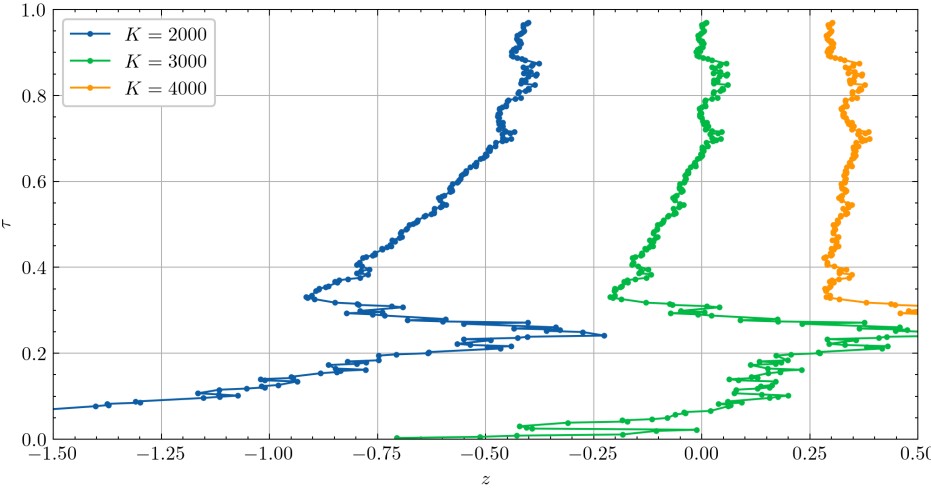

Figure 9: S&P 500 Put option paths, expiry on 19.06.2020. The path is depicted from 01.06.2019 to 01.06.2020.

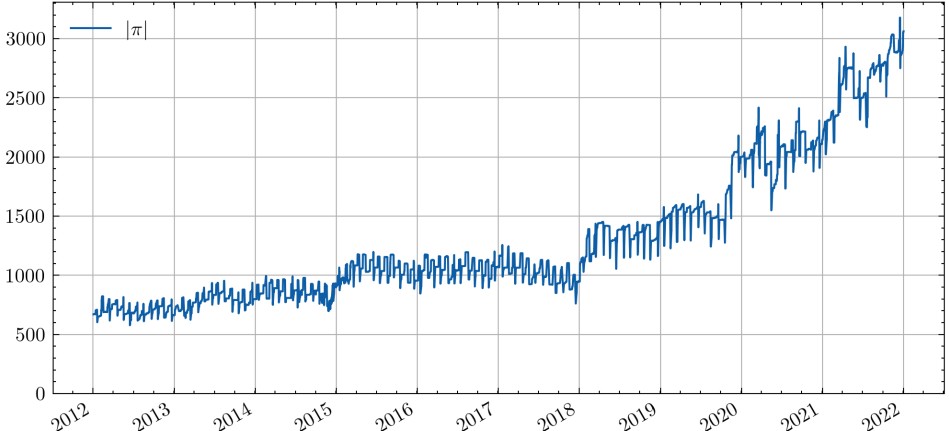

Figure 10: Development of the total number of listed Put and Call options (out-the-money and contained in our smoothing domain).

## C.9    ANALYSIS OF COMPUTATIONAL COMPLEXITY

Let $\pi_{\text{in}}$ be the $(\tau, k)$-coordinates of the volatility inputs $\mathbf{v}$ and $\pi_{\text{out}}$ the points at which the smoothed surface is to be computed. As part of the forward pass, the computation of the kernel integrals in each layer involves for each $y \in \pi_{\text{in}} \cup \pi_{\text{out}}$ the summation of $|\mathcal{N}(y)| \leq K$ many terms. In total, this amounts to the summation of $O((n_{\text{in}} + n_{\text{out}})K)$ many vectors, where $n_{\text{in}} = |\pi_{\text{in}}|$ is the number of input datapoints and $n_{\text{out}} = |\pi_{\text{out}}|$ is the number of queried output points. The production of each of these vectors involves the evaluation of the kernel FNNs as well as the matrix multiplication of its output with the previous hidden state $\tilde{h}_j(x)$ (in our implementation, a $16 \times 16$ matrix-vector multiplication). We take these as base units of computation and do not decompose their complexity further (in terms of e.g. the hidden channel size). Moreover, each layer has its own lifting network (the last one has a projection as well) and a local linear transformation (the first one has not), which involve $O(n_{\text{in}} + n_{\text{out}})$ many FNN evaluations/matrix multiplications. The computational effort therefore is $O(J(n_{\text{in}} + n_{\text{out}})K)$ (where $J$ is the number of layers), counted in evaluations of (relatively small) feedforward neural networks. We derive the following strategies to control the "time-to-smoothed-surface":

- Changing the subsample size $K$: We refer to Appendix C.6, which contains an ablation study for $K$. It shows that there is potentially quite a bit of leeway to reduce $K$ below the level of $K = 50$, reducing the complexity of the forward pass.

- Decreasing the resolution of the output surface: Lower-resolution output can be traded for faster execution speed.

- Targeting the constants: It could be motivated to investigate smaller FNN components for kernels, liftings, and projections with faster execution speeds.

- Accelerate compute: Since our method is purely compute-based, the use of more and/or faster GPUs will allow to process surfaces more quickly. This is a great advantage over instance-by-instance smoothing, whose fitting routines are sequential in nature and hard to accelerate.

We benchmarked the execution with $n_{\text{in}} = 897$ and $n_{\text{out}} = 2500$ (i.e., with an output grid of size $50 \times 50$) on a consumer-grade laptop CPU as well as an Nvidia Quadro RTX 6000 GPU in Table 10. The numbers confirm the linear scaling of compute in the subsample size $K$ and the great accelerability of our method using GPUs. They also show that computation times for typical grid sizes are fast enough for performing smoothing several times per second, which is more than sufficient for most applications. However, there exist participants in modern financial markets that operate at sub-millisecond timescales (*high-frequency trading* or *HFT*). At the same time, our current implementation prioritizes methodological validation over speed optimization (in particular the graph construction is brute-forced and takes around 100 ms), and prevents us from drawing definitive conclusions about the method's applicability across the full spectrum of HFT applications. A

comprehensive optimization study targeting HFT-specific requirements lies beyond the scope of this work.

Table 10: Computation times for different subsample sizes $K$ on CPU and GPU.

| $K$ | 10 | 20 | 30 | 40 | 50 |
|---|---|---|---|---|---|
| CPU (ms) | 74.6 | 149 | 215 | 273 | 336 |
| GPU (ms) | 3.57 | 6.93 | 10.0 | 13.2 | 16.7 |

# D    ADDITIONAL PLOTS

## D.1    EXAMPLE PLOTS

We plot the results of operator deep smoothing (OpDS) vs. SVI on example inputs. To aid the visual clarity of our plots, we display only every third market datapoint.

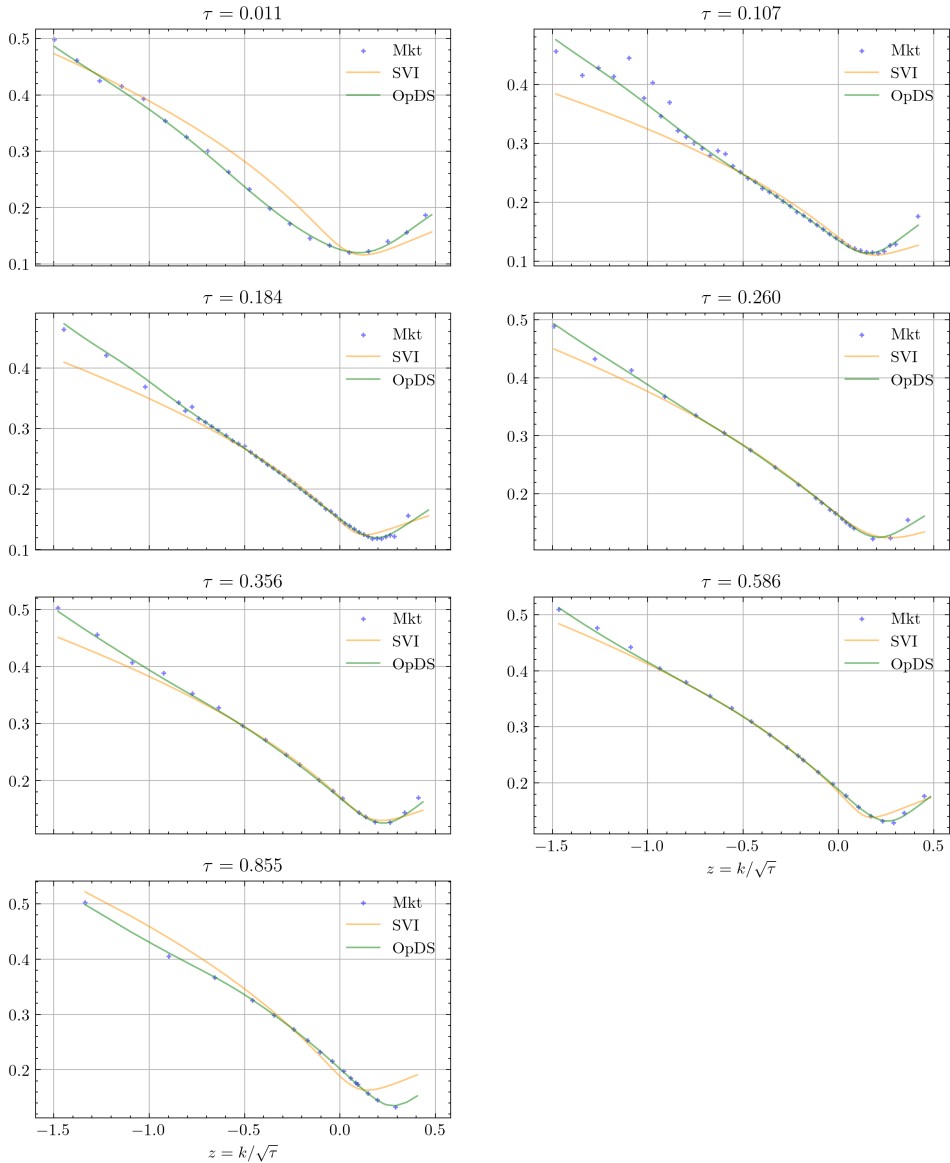

Figure 11: Smoothing of quotes $\mathbf{v} \in \mathcal{D}_{\text{val}}$ from 20.07.2012 at 10:50:00.

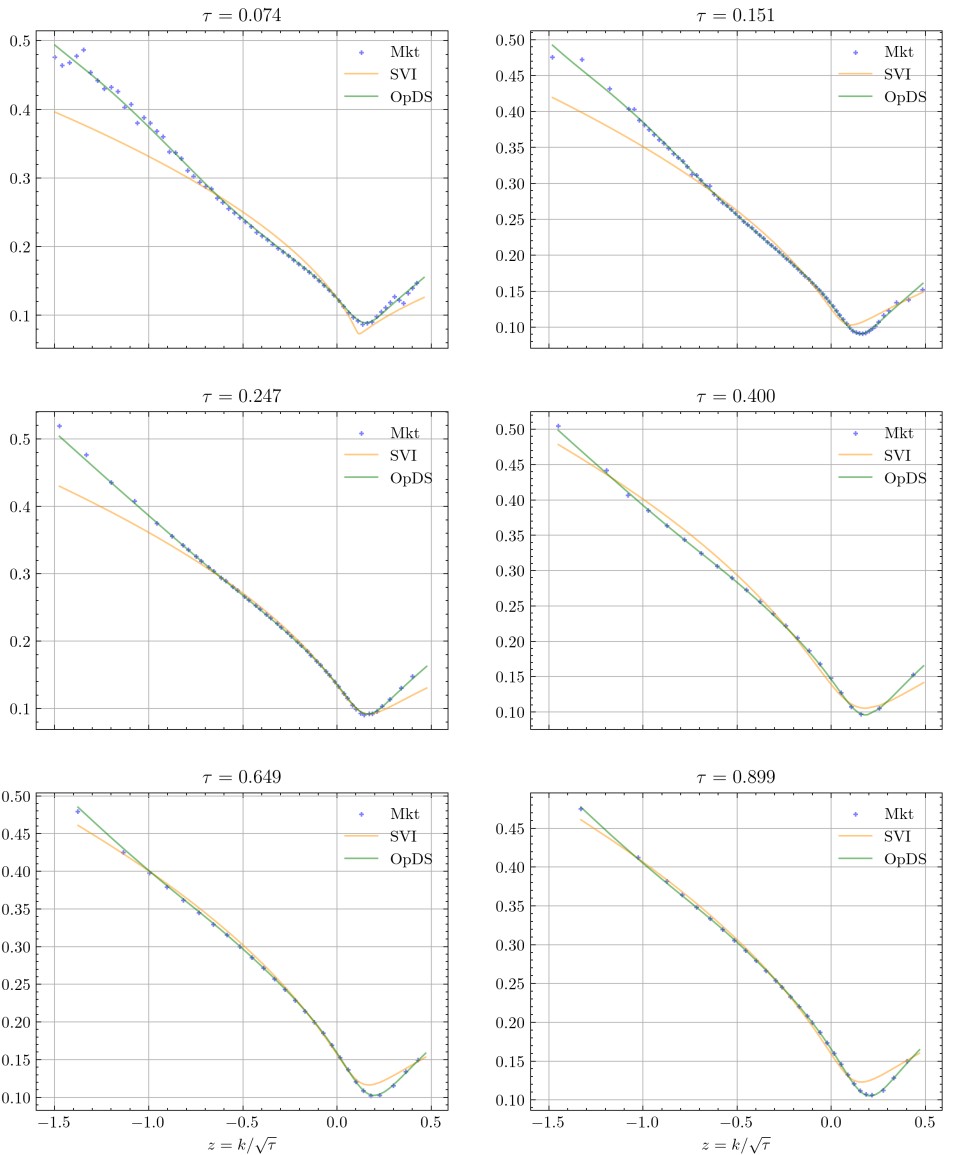

Figure 12: Smoothing of quotes $\mathbf{v} \in \mathcal{D}_{\text{val}}$ from 21.10.2016 at 13:10:00.

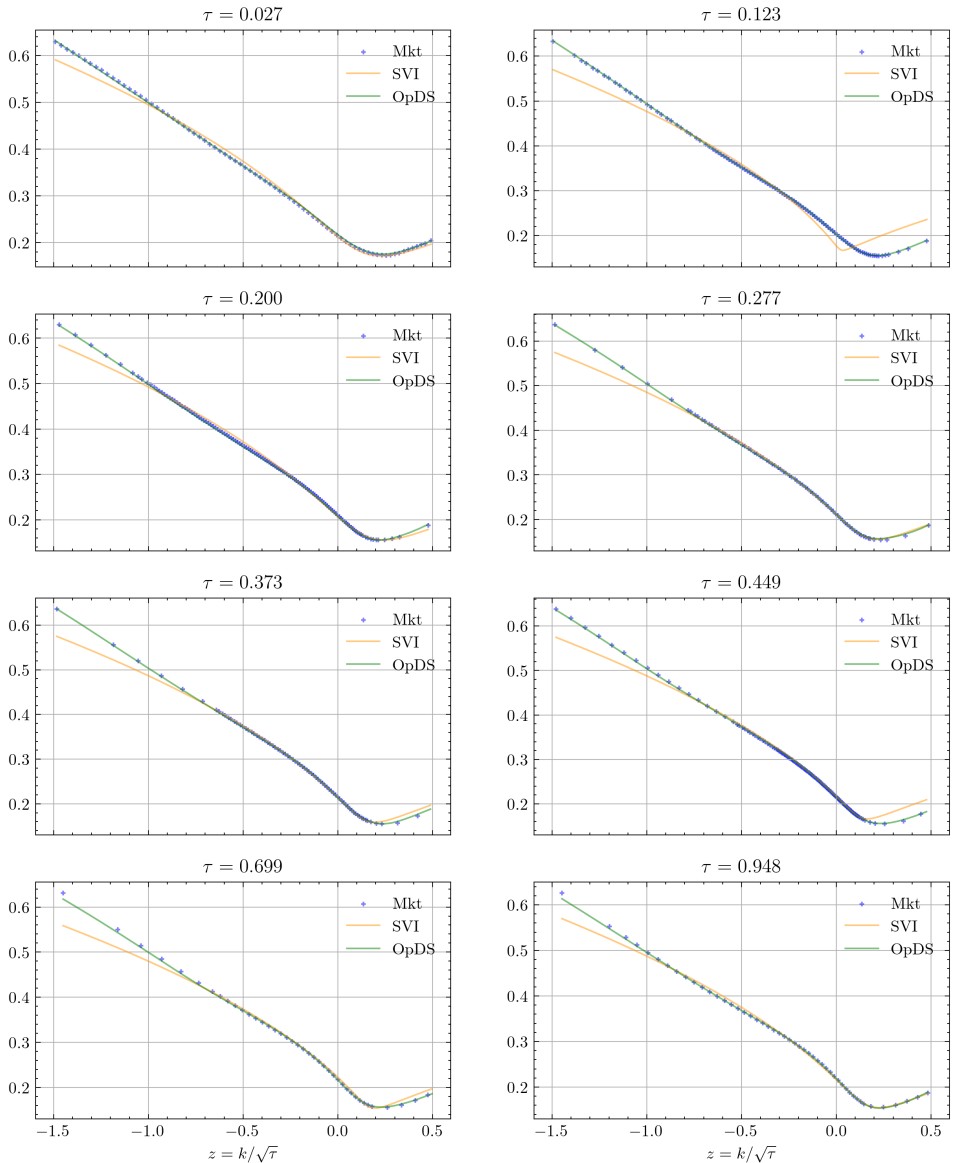

Figure 13: Smoothing of quotes $\mathbf{v} \in \mathcal{D}_{\text{train}}$ from 04.01.2021 at 10:50:00.

### D.2 MONTHLY BREAKDOWN OF SPATIAL DISTRIBUTIONS OF BENCHMARK METRICS ON TEST DATASET

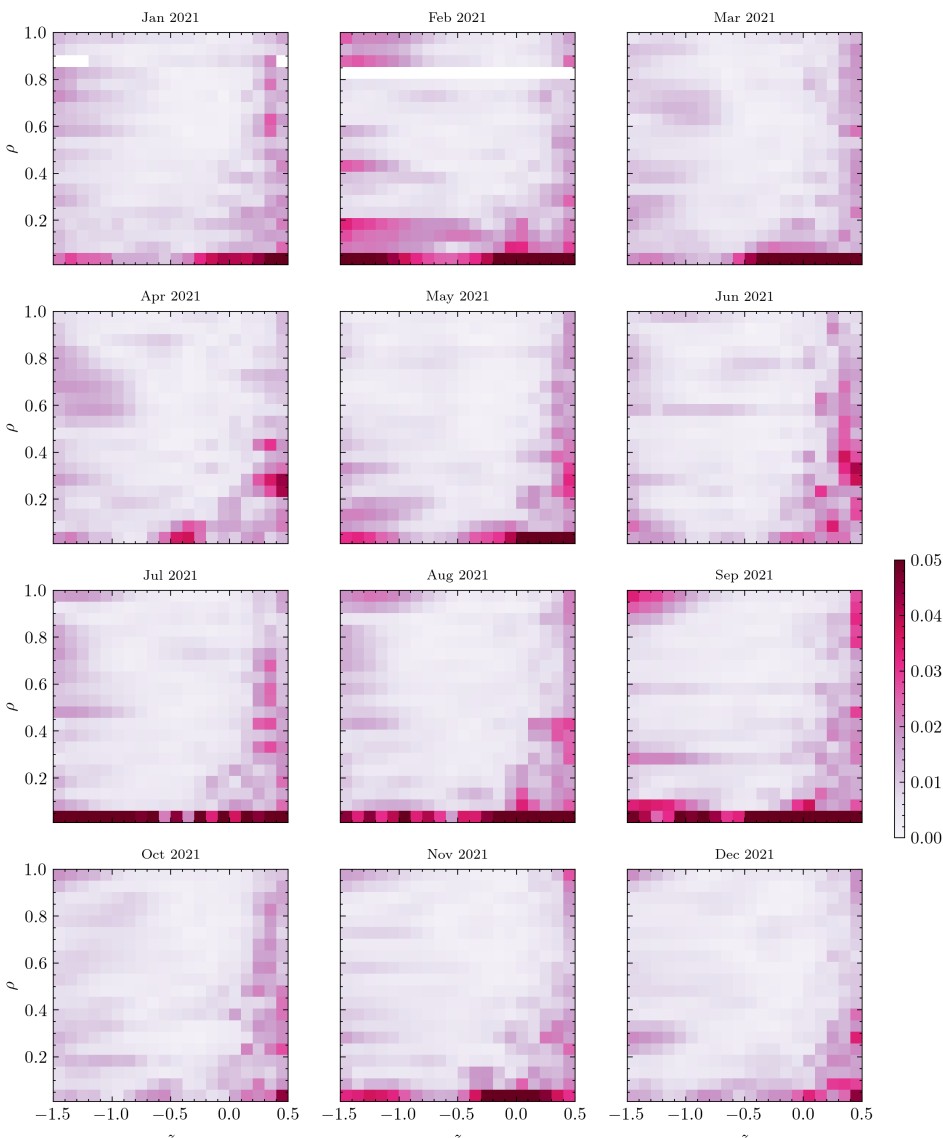

Figure 14: Average spatial distribution of $\delta_{\text{abs}}$ on $\mathcal{D}_{\text{test}}$ for OpDS with monthly finetuning, per month. Blank cells indicate that no data was available for the particular region in the respective month.

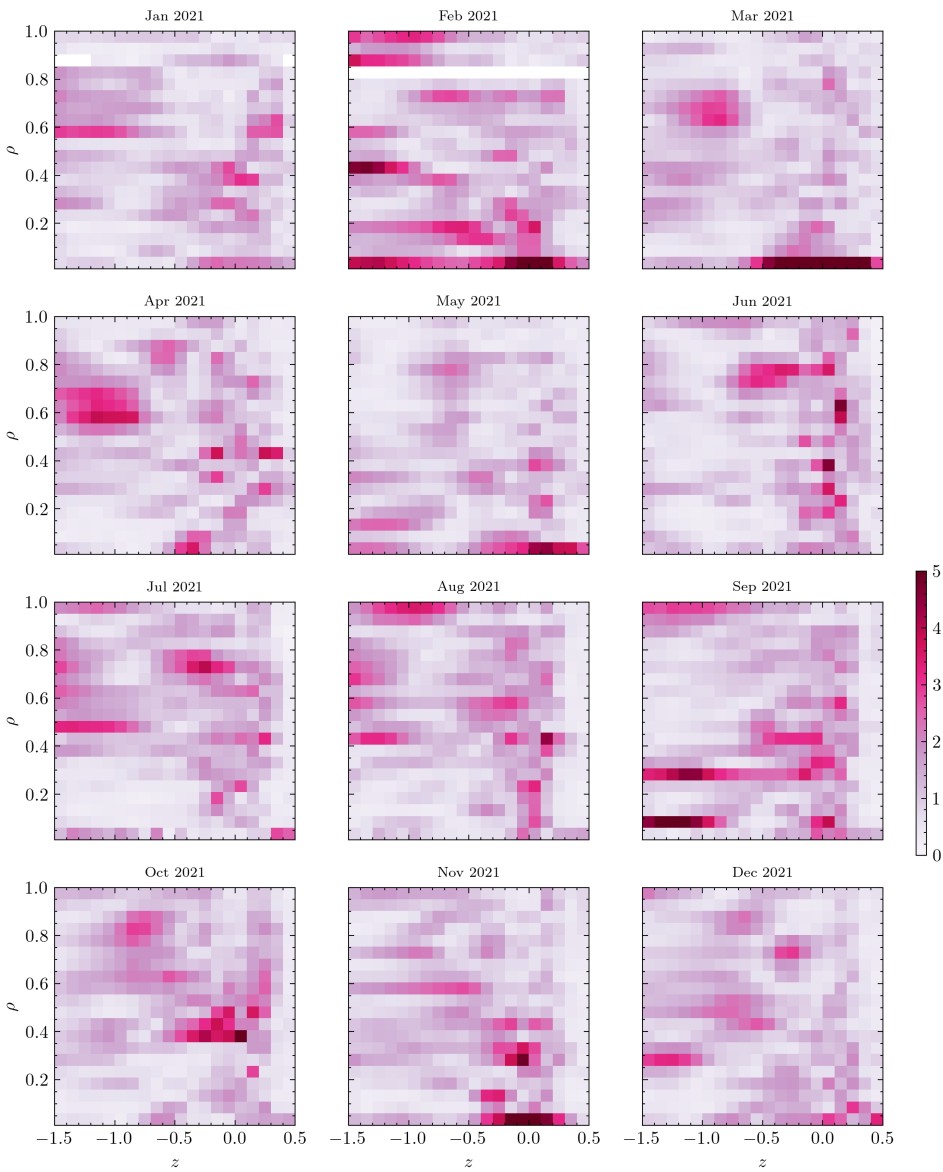

Figure 15: Average spatial distribution of $\delta_{\text{fit}}$ on $\mathcal{D}_{\text{test}}$ for OpDS with monthly finetuning, per month. Blank cells indicate that no data was available for the particular region in the respective month.

### D.3 EXAMPLE PLOTS: OPTION METRICS END-OF-DAY US INDEX OPTIONS DATA

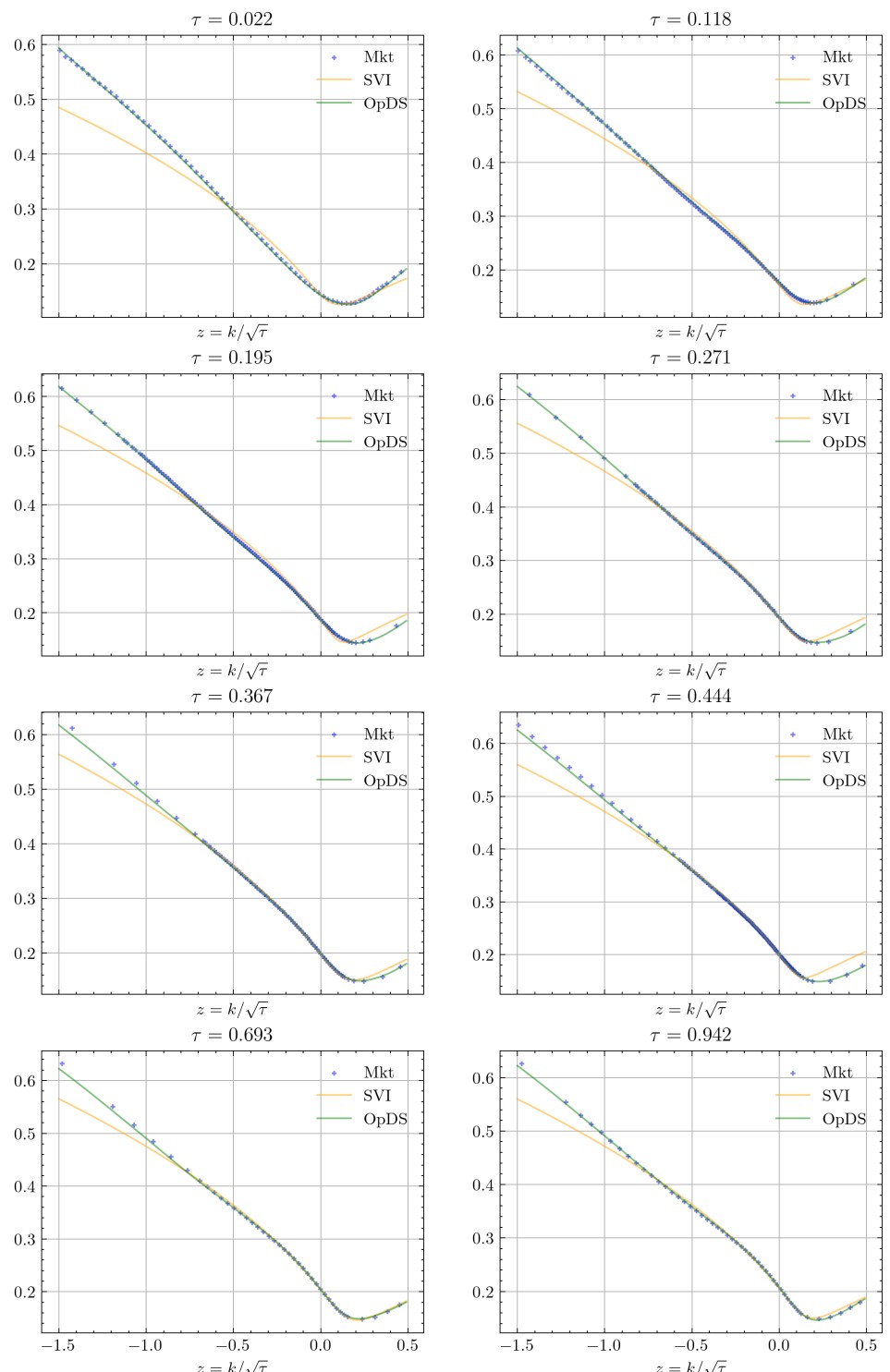

Figure 16: Smoothing of SPX end-of-day data from 07.01.2021. Every third datapoint displayed.

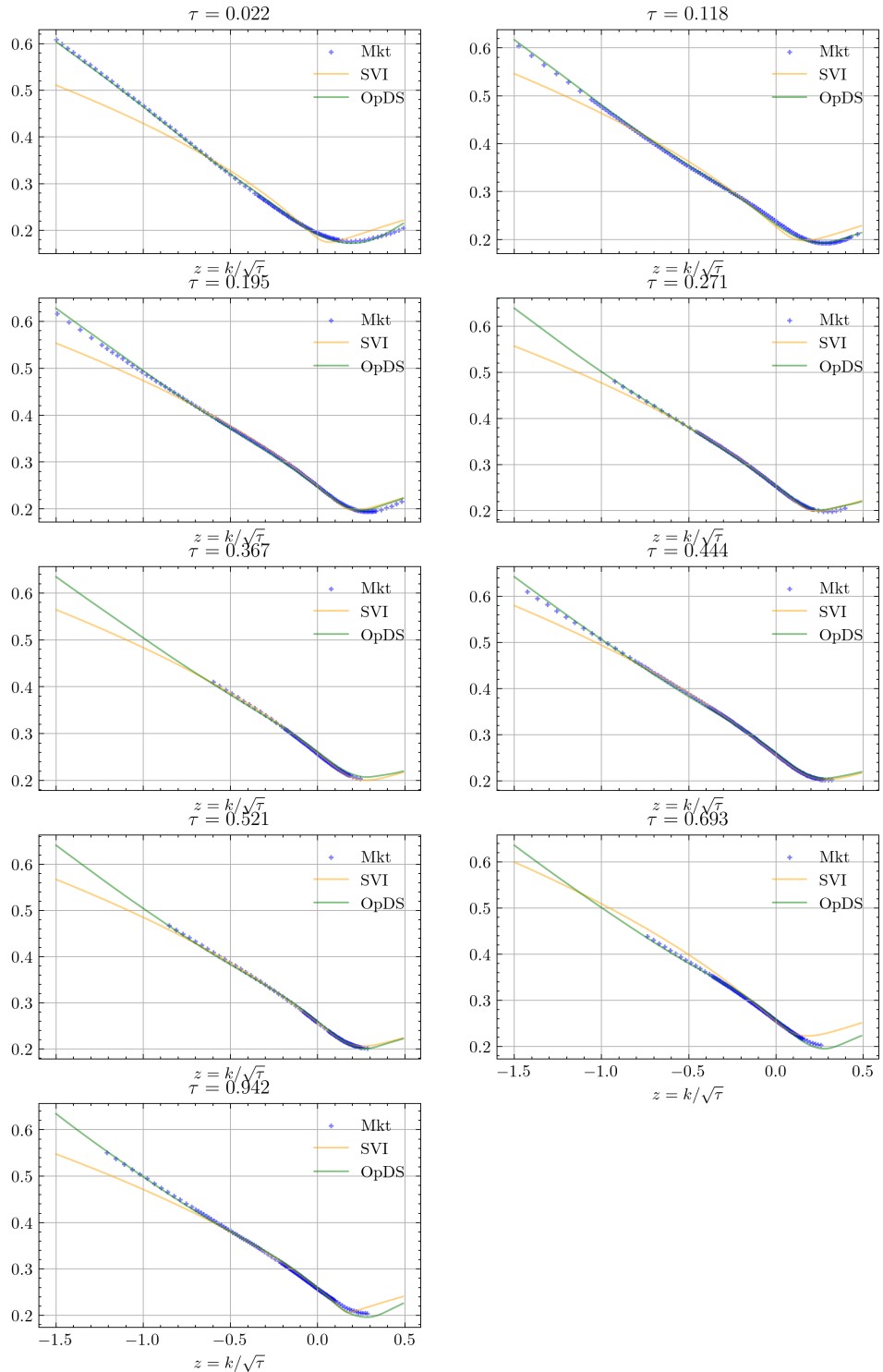

Figure 17: Smoothing of NDX end-of-day data from 07.01.2021. Every second datapoint displayed.

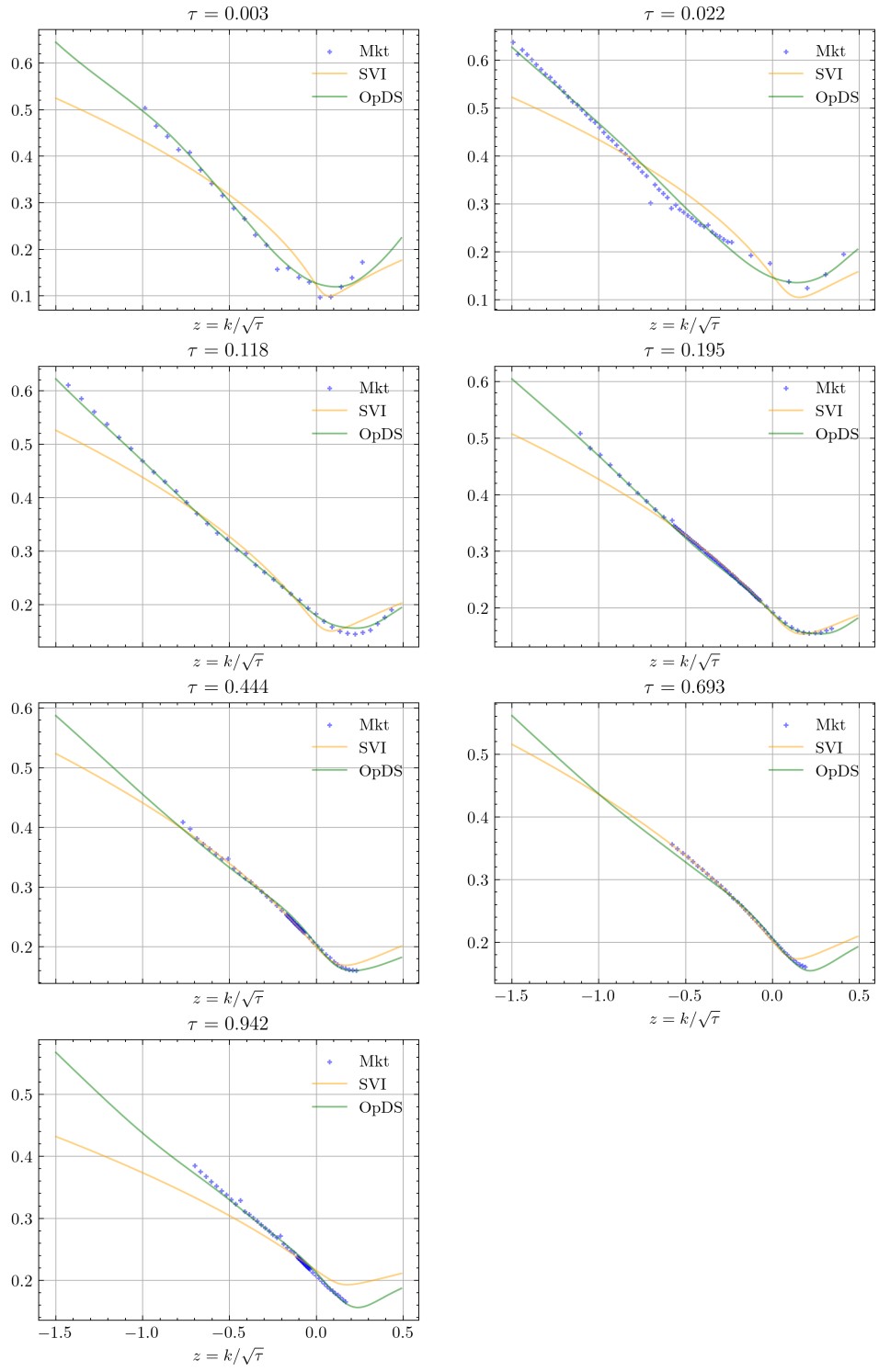

Figure 18: Smoothing of DJX end-of-day data from 07.01.2021. Every datapoint displayed.

### D.4 EXAMPLE PLOTS: SYNTHETIC SSVI DATA

*SSVI* is a continuous interpolation scheme for SVI (see Section E) introduced in Gatheral & Jacquier (2014). We use to generate synthetic volatility surface data, to which we then apply our smoothing method. We consider the following formulation of SSVI (taken from Lind & Gatheral (2023)):

$$\hat{v}_{\text{SSVI}}^{\phi}(\tau, k) = \sqrt{\frac{\theta_\tau}{2\tau}\left(1 + \rho\varphi(\theta_\tau)k + \sqrt{(\varphi(\theta_\tau)k + \rho)^2 + (1 - \rho^2)}\right)},$$

where $\varphi(\theta_\tau) = \eta\theta_\tau^\gamma$ and

$$\theta_\tau = \theta\,\tau + (V - \theta)\frac{1 - e^{-\kappa_1 T}}{\kappa_1} + (V' - \theta)\frac{\kappa_1}{\kappa_1 - \kappa_2}\left(\frac{1 - e^{-\kappa_2 T}}{\kappa_2} - \frac{1 - e^{-\kappa_1 T}}{\kappa_1}\right),$$

making $\phi = (V, V', \theta, \rho, p, \eta, \gamma, \kappa_1, \kappa_2)$ the parameter vector. We set the parameters to the following values, reported as historical averages for the S&P 500 over the period 1996-2021 in Lind & Gatheral (2023):

| $V$ | $V'$ | $\theta$ | $\rho$ | $p$ | $\eta$ | $\gamma$ | $\kappa_1$ | $\kappa_2$ |
|------|------|------|------|------|------|------|------|------|
| 0.04 | 0.04 | 0.11 | $-0.5$ | 0.01 | 1.19 | 0.49 | 5.5 | 0.1 |

We produce synthetic market data $\mathbf{v}_{\text{SSVI}}$ by evaluating the SSVI instance $\hat{v}_{\text{SSVI}}^{\phi}$ on the synthetic $(8 \times 51)$-grid

$$\pi_{\text{in}} = \pi_\rho \times \pi_z$$

with $\pi_\rho = \{0.16, 0.28, 0.4, 0.52, 0.64, 0.76, 0.88, 1\}$ and $\pi_z = \{-1.5, -1.46, \ldots, 0.46, 0.5\}$. We then proceed to smooth $\mathbf{v}_{\text{SSVI}}$ with our method, producing $\hat{v}_{\text{OpDS}} = \tilde{F}^\theta(\mathbf{v}_{\text{SSVI}})$, and plot the results in Figure 19. Moreover, we compute the MAPE smoothing error, both over the grid $\pi_{\text{in}}$, as well as over a high-resolution $(100 \times 100)$-grid

$$\pi_{\text{out}} = \pi_\rho \times \pi_z,$$

where $\pi_\rho$ and $\pi_z$ discretize $[0.01, 1]$ and $[-1.5, 0.5]$, respectively, uniformly with one hundred nodes each. We achieve the following values:

| $\pi_{\text{in}}$ | $\pi_{\text{out}}$ |
|------|------|
| 2.01% | 2.42% |

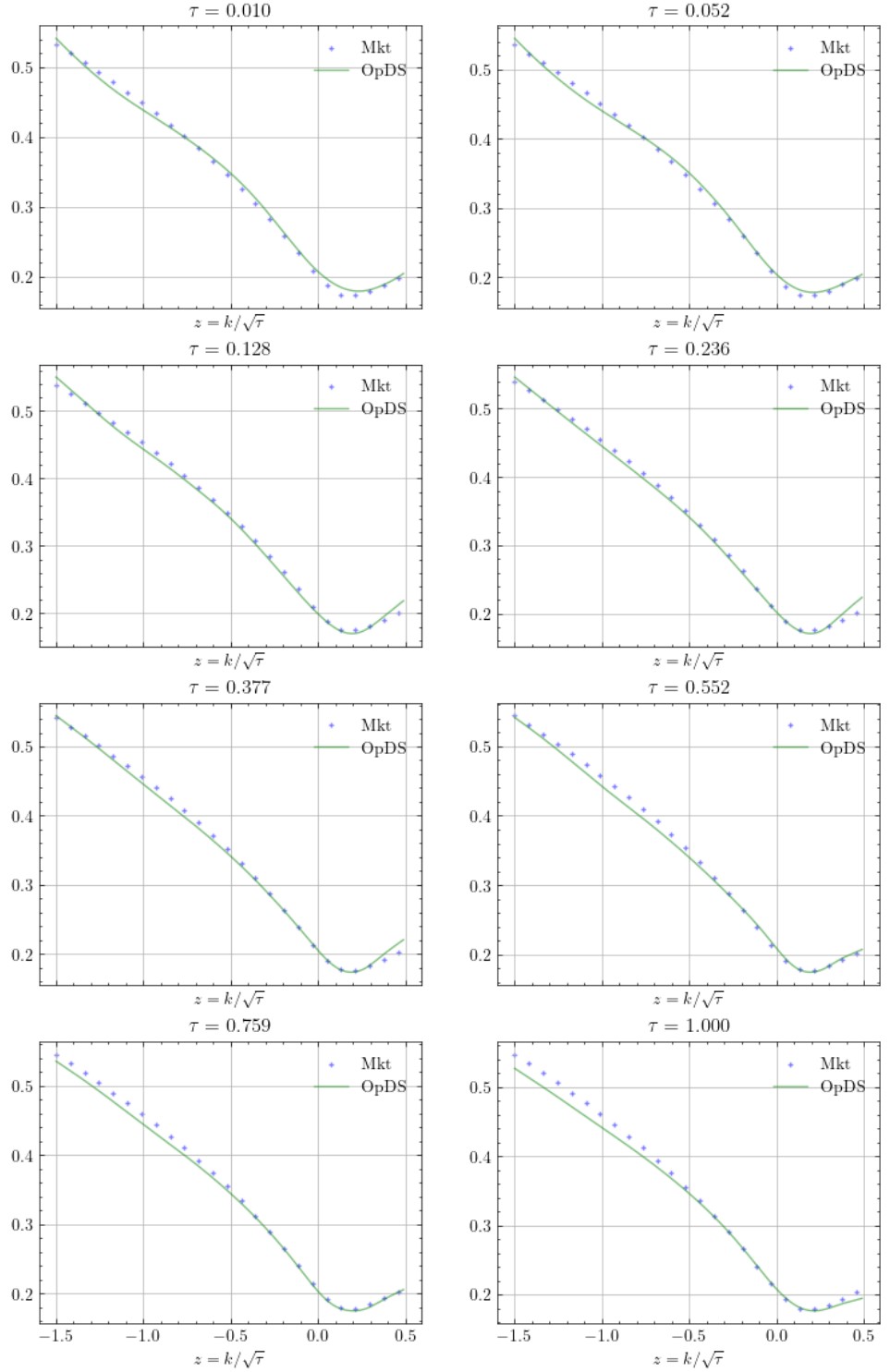

Figure 19: Smoothing of synthetic SSVI data. Every second datapoint displayed.

# E   SVI

SVI, originally devised by Gatheral (Gatheral, 2004), stands for *Stochastic Volatility Inspired* and is a low-dimensional parametrization for implied volatility slices (namely for each maturity). It captures the key features of implied volatility of Equity indices, and has become the industry-wide benchmark for implied volatility smoothing on such markets. Its *raw* variant parametrizes the "slice" of implied volatility at a given time-to-expiry $\tau$ as

$$\hat{v}_\tau^\phi(k) = \sqrt{\frac{a + b\left(\rho(k-m) + \sqrt{(k-m)^2 + \sigma^2}\right)}{\tau}}, \quad k \in \mathbb{R},$$

where $\phi = (a, b, \rho, m, \sigma)$ is the parameter vector.

**Calibration**   While stylistically accurate, SVI does not easily guarantee absence of static arbitrage opportunities, and several authors have investigated this issue (Gatheral & Jacquier, 2014; Martini & Mingone, 2022; Mingone, 2022; Martini & Mingone, 2023). To produce our SVI benchmark we therefore rely on the constrained SLSQP optimizer provided by the *SciPy* scientific computing package for Python, with the mean square error objective, a positivity constraint and the constraint equation 2 (computed in closed form), and the following parameter bounds:

$$a \in \mathbb{R}, \quad b \in [0, 1], \quad \rho \in [-1, 1], \quad m \in [-1.5, 0.5], \quad \sigma \in [10^{-8}, 2].$$

We ignore the calendar arbitrage condition.

