# OpenReview forum: "Operator Deep Smoothing for Implied Volatility"
_ICLR.cc/2025/Conference — ICLR 2025 Poster_

### Official Review · Reviewer_Xa1Y · 2024-11-04

**Soundness:** 3
**Presentation:** 3
**Contribution:** 3
**Rating:** 6
**Confidence:** 2

**Summary:**

**Disclaimer**: I have no background on financial engineering so I cannot comment on the originality/significance aspect of this work. I might have missed essential points and only base my review on technical soundness and empirical validation

This paper introduces a method to nowcast implied volatility based on neural operators. This is achieved by a graph neural operator with MLPs as the lifting/projection layers and the kernel functions. The model uses a composite loss function that includes the fitting loss, no-arbitrage constraints, enforcing monotonicity and regularization terms to ensure smoothness. The method improves over the baseline SVI model.

**Strengths:**

The proposed method appears to be an interesting approach to model implied volatility surfaces for option pricing. The method through composite loss functions that include constraints such as no-arbitrage allows for direct training of a neural network architecture that directly provides smooth surfaces that directly satisfy these constraints. While I cannot comment directly how difficult/hands-on other alternative approaches are in this domain, the proposed model appears to have the flexibility to introduce other loss terms or vary the graph structure to incorporate other structural properties directly into the model. The model leads to lower errors compared to the SVI method.

**Weaknesses:**

The paper gives a good detail on the exact instantiation and implementation of the proposed method. I understand that there might be little prior work to base this paper on. However, it is unclear how the exact choices have been made and what would be their impact if changed. Note that the choices here might be purely made by domain knowledge, which I cannot judge and therefore might miss the specific considerations made here. However, I think the reader would benefit from more detail on the following:

**In-Neighborhood sets**: The choice of in-heighborhood sets is arguably key model choice in this work. However, this work lacks detail on how the choice has been made and the subsampling heuristic has been devised. The authors argue that the choice is made because volatility smoothing requires limited global information exchange. I would then expect that the error of this method would go up if larger neighborhood sets are used. This paper would benefit from an additional analysis on changing the $\bar{p}$ and $K$ parameters.

**Other model choices**: Again, there is little information about all the other model choices and/or instantiations. In particular, the choice of the kernel functions as MLP (as opposed to other choices) would be interesting for the reader. I'm also wondering whether the MLP can output negative values? I understand that there is a softplus operation in the architecture and ensures positive values but again the choice of a MLP would benefit from further explanation/ablation.

**Loss function weighting coefficients:** I would kindly ask the authors to include more detail how the weighting of the loss function has been performed. As it is stated right now, there is no information on how to select/adjust the weightings on a new pr.oblem. It would help the dissemination the method if the authors would share their methodology.

**Ablations**: The claims in the paper could be strengthened by providing ablations In particular, different (wider) choices of the graph neighborhood would justify the choice of the authors. Additionally, omitting the no-arbitrage term and any of the other auxiliary terms would give more insight on the effect of the individual chosen components.

I would reconsider my score if additional justification on the specific choices are included in the paper and the additional ablations on in-neighborhood and loss function weighting are performed.

**Questions:**

See Weaknesses section.

---

> ### Author Response · Authors · 2024-11-21
>
> Dear Reviewer Xa1Y,
>
> we thank you for your review and for the clear suggestions of how to improve the paper.
> We reworked the relevant sections and included ablation studies, aiming at improving the explanation of our architectural choices and validating them.
> Please have a look at revised PDF and our summary below.
>
>
> ### In-Neighborhood Sets
>
> You mention:
> > The choice of in-neighborhood sets is arguably key model choice in this work. [...] This paper would benefit from an additional analysis on changing the $\bar{p}$ and $K$ parameters.
>
> To address this point, we added the new paragraph "The Choice of In-Neighborhoods" in Appendix C.2 (as well as an ablation study in Appendix C.6, see below).
> In it, we expand on the following points:
> - The truncation of the in-neighborhoods in $\rho$-direction by $\bar{\rho} = 0.3$ is motivated by concrete domain-specific considerations: The maximum distance between option slices (for our data), is around 0.269. Our choice then ensures slices to be able "communicate" with each other in pairs of two, which is enough to enforce the monotonicity constraint of Theorem 2.1 of the paper. On the other hand, it is not motivated to increase $\bar{\rho}$ above this level, since our deep architecture allows information to propagate over the whole length of the domain in the $\rho$-direction (but also we wouldn't expect the error to go up when doing so – we hypothesize the network will pick the relevant information even from a larger neighborhood; see the next point). Moreover, an ablation study for $\bar{\rho}$ would require retraining the whole GNO from scratch, which time-/resource-constraints forbid us at this point.
> - Our neighborhood sets are not limited in size in the $z$-direction; this is a defensive choice that, in particular (relating to previous point), shows that the GNO is able to extract the relevant information even from an untruncated in-neighborhood (along $z$). Moreover, this choice readies the model to arbitrarily extrapolate in $z$-direction.
> - We identified the subsample-size $K=50$ by manual experimentation. Prompted by your remarks, we performed an ablation study for different values of $K$, discussing the interesting results in newly introduced Appendix C.6 (adding Tables 6 and 7). The results ultimately validate our choice.
> - We add a description of our subsampling heuristic instead of solely referring to the code material.
>
>
> ### Loss Function Weighting
>
> You mention:
> > I would kindly ask the authors to include more detail how the weighting of the loss function has been performed. As it is stated right now, there is no information on how to select/adjust the weightings on a new problem.
>
> We reported in Appendix C.3 that we retrieved our weighting of the loss terms by manual experimentation, led by the findings of Ackerer et al. (2020).
> Indeed, this has allowed us to produce effectively arbitrage-free results.
> Prompted by your remark, we performed an ablation study for the weight of the arbitrage penalty terms and discuss its results in newly introduced Appendix C.7 (adding Tables 8 and 9).
>
>
> ### Other model choices
>
> You mention:
> > Again, there is little information about all the other model choices and/or instantiations. In particular, the choice of the kernel functions as MLP (as opposed to other choices) would be interesting for the reader. I'm also wondering whether the MLP can output negative values? [...] the choice of a MLP would benefit from further explanation/ablation.
>
> We use MLPs for all remaining components of the GNO to keep things simple:
> The universal approximation theorem for neural operators in Kovachki et al. (2023) is based on using plain MLPs as well, relying on their universality properties as approximators between finite-dimensional spaces.
> While we agree that it would be interesting to investigate the effectiveness of different architectures in this context, it has been our strategy to stick to straightforward GNO-components for this paper.
> As for the configuration of the MLPs, we used manual experimentation led by conventional wisdom:
> For example, for the kernel-MLP, we chose the number of nodes of the hidden layers at the power of two closest to three times the number of nodes of the input layer, and we picked GELU as hidden activation for its smoothness properties, and use no output activation since the kernel parametrizes a linear transformation and definitely should be able to take negative values (answering your question).
>
>
> Thank you very much again for your review which we believe has helped us substantially improve the transparent presentation of our methodology (and it is reassuring to see our model choices validated by the ablation studies you suggested).

---

> > ### Comment · Reviewer_Xa1Y · 2024-11-22
> >
> > Thank you for the additional ablations. I have increased my score.

---

> > > ### Author Response · Authors · 2024-11-25
> > >
> > > Thank you very much.
> > > We wanted to mention that we added Figure 5 into the paper, which details the in-neighborhood construction visually.

---

### Official Review · Reviewer_o4xg · 2024-11-09

**Soundness:** 3
**Presentation:** 2
**Contribution:** 2
**Rating:** 6
**Confidence:** 2

**Summary:**

This paper proposes a novel approach to nowcast implied volatility using neural operators, advancing the technique of implied volatility smoothing by constructing consistent, smooth surfaces that reflect current option market prices. Unlike traditional machine learning models, which struggle with dynamically changing spatial configurations in option price data, this work leverages a graph neural operator architecture to achieve robust and accurate predictions. Extensive benchmarks showcase that this work achieves superior generalization and accuracy over conventional neural networks and the SVI parametrization.

**Strengths:**

1. This paper is well-written and easy to follow.

2. This proposed method is resilient to input subsampling, aligns with no-arbitrage conditions, and eliminates the need for extensive data pre-processing.

**Weaknesses:**

1. The contribution is limited. It is focused on applying the Graph Neural Operator (GNO) architecture to the specific task of nowcasting implied volatility, without modifications to the operator itself.

2. It's important to discuss various neural operators and clarify why the Graph Neural Operator (GNO) was chosen for this task.

**Questions:**

please refer to the weaknesses part

---

> ### Author Response · Authors · 2024-11-13
>
> Dear Reviewer o4xg,
>
> thank you very much for your review.
> It is encouraging that you find our paper well-written and easy to follow, and that you agree with the strengths of our method with respect to input subsampling, no-arbitrage conditions, and elimination of data pre-processing.
>
> As for mentioned weaknesses, however, we believe there might have been an oversight.
> You mention:
> > 1. The contribution is limited. It is focused on applying the Graph Neural Operator (GNO) architecture to the specific task of nowcasting implied volatility, without modifications to the operator itself.
> > 2. It's important to discuss various neural operators and clarify why the Graph Neural Operator (GNO) was chosen for this task.
>
> However, our employed graph neural operator is, in fact, modified compared to the conventional version, and we devote the paragraph "Interpolating Graph Neural Operator" in Section 3.2 to this (as well as Appendix B).
> More precisely, our modification consists of dropping the local linear transformation in the first layer and careful construction of the graph structure to enable data interpolation.
> We state:
>
> > We therefore propose a new architecture for operator deep smoothing leveraging GNOs’ unique ability to handle irregular mesh geometries. We use a purely non-local first layer (dropping the pointwise linear transformation), and use it to produce hidden state at all required output locations, enabling subsequent layers to retain their local transformations. (Page 7 of the paper.)
>
> While our tweak to the classical GNO architecture is subtle, it is crucially important to enable its use for interpolation tasks.
> We explain --and argue that this moreover addresses your second mentioned weakness--:
> > Various neural operator architectures exist, mostly arising from the kernel integral transform framework of Kovachki et al. (2023). Most prominently, these include Fourier neural operators (FNO), delivering state-of-the-art results on fixed grid data, as well as graph neural operators (GNO), able to handle arbitrary mesh geometries (both reviewed in Appendix A.1). While highly effective with documented universality, these neural operators are not directly applicable for interpolation tasks as their layers include a pointwise-applied linear transformation, which limits the output to the set of the input data locations. (Page 7 of the paper.)
>
> Finally, we want to reiterate on your claim that our contribution "is focused on applying the GNO architecture to the specific task of nowcasting implied volatility".
> However, the entire Section 3.2 is kept general and not specific to implied volatility.
> Instead, we introduce the concept of *operator deep smoothing* as the generally applicable idea of using discretization-invariant neural operators approximating the identity operator to solve irregular interpolation tasks and explain the required modifications necessary to current neural operator architectures.
> Our practical experiments and much of the rest of the paper then indeed remain limited to implied volatility smoothing, which however is a massively important task in the financial industry, and which we believe benefits dramatically from our foundational, data-focused operator approach and is highly suited to showcase its unique advantages.
>
> We hope the above explanations helped to dispel your doubts regarding our contribution.
> Please do let us know if any unclarities or suggestions for improvement remain.
> For example, we could expand our review of existing neural operator architectures in Appendix A, to support our argumentation in Section 3.2.
>
> Thank you very much!

---

> > ### Comment · Reviewer_o4xg · 2024-11-24
> >
> > Dear Authors,
> >
> > Thank you for your clarification. I have reconsidered your response and raised my scores accordingly.

---

### Official Review · Reviewer_GefV · 2024-11-10

**Soundness:** 3
**Presentation:** 3
**Contribution:** 2
**Rating:** 6
**Confidence:** 4

**Summary:**

This paper introduces a novel approach to smoothing implied volatility using neural operators, focusing on Graph Neural Operator (GNO) architectures. Unlike traditional parametric models or classical neural networks, the proposed operator deep smoothing method directly maps observed implied volatilities to smooth, arbitrage-free surfaces, eliminating the need for instance-by-instance recalibration. By harnessing the unique capability of neural operators to handle data of varying sizes and spatial arrangements, the model effectively addresses the challenges posed by the dynamic and irregular nature of real-world financial data. Validated on a decade of intraday S&P 500 options data, the method outperforms the SVI benchmark and other machine learning techniques while generalizing effectively to unseen datasets, including end-of-day options for other indices, without retraining.

**Strengths:**

- **Novelty:** This paper proposes a novel application of GNO architectures to implied volatility smoothing, a task traditionally reliant on parametric models like SVI. By leveraging the discretization-invariance of neural operators, the method effectively addresses the challenges of dynamic and irregular financial data, marking a significant step forward in financial engineering.

- **Practical Significance:** The operator deep smoothing approach eliminates the need for instance-by-instance recalibration, streamlining the online calibration process. This drastically reduces computational overhead, making the method highly practical for real-time applications in trading, risk management, and other financial operations.

- **Robust Validation:** The method is extensively validated on a decade of intraday S&P 500 options data, demonstrating superior accuracy compared to the SVI benchmark and other machine learning techniques. Its strong generalization performance on unseen datasets, including end-of-day options for other indices, further highlights its robustness and adaptability.

 - **Reproducibility:** The authors provide open-source code, model weights, and detailed implementation details, including architecture, loss functions, and training setups. These resources ensure the experiments are easy to replicate and extend.

**Weaknesses:**

- **Limited Benchmark Comparisons:** The paper benchmarks its approach against SVI [1] and Ackerer et al. [2] but does not include comparisons with other key methods, such as SSVI [3] and VAE-based approaches [4]. Incorporating these would provide a more comprehensive evaluation. Additionally, using synthetic data, as in [2], could further strengthen the experimental validation.

 - **Insufficient Analysis of Computational Efficiency:** While the paper highlights the elimination of instance-by-instance recalibration, it lacks a detailed discussion on computational complexity, including runtime and memory requirements. These metrics are crucial for assessing the method’s practicality in high-frequency financial contexts.

 - **Abstract Treatment of Discretization-Invariance:** The explanation of discretization-invariance is largely theoretical. Providing concrete financial scenarios, such as handling abrupt market shifts, would better illustrate its practical significance and strengthen the narrative.

- **Clarity of Experimental Figures**: Figures 3 and 4 have unclear legends and insufficient annotations, which reduce their interpretability. Improving their clarity and labeling would make the results more accessible and impactful.

[1] A parsimonious arbitrage-free implied volatility parameterization with application to the valuation of volatility derivatives.

[2] Deep Smoothing of the Implied Volatility Surface.

[3] Arbitrage-free SVI volatility surfaces.

[4] Variational Autoencoders: A Hands-Off Approach to Volatility.

**Questions:**

See Weaknesses.

---

> ### Author Response · Authors · 2024-11-25
>
> Dear Reviewer GefV,
>
> thank you very much for your insightful review, including an exhaustive summary of the strengths of our work.
> As for your mentioned weaknesses, they have prompted us to rework portions of our paper and include additional material which we believe has improved and rounded off its presentation.
> We address in more detail below.
>
> ### Abstract Treatment of Discretization-Invariance
>
> You mention:
> > The explanation of discretization-invariance is largely theoretical. Providing concrete financial scenarios, such as handling abrupt market shifts, would better illustrate its practical significance and strengthen the narrative.
>
> Thank you for pointing this out.
> We have newly introduced Appendix C.8 with two new plots detailing the spatial and dimensional irregularity of observed options data.
> It includes Figure 9, which plots the time-to-expiry/log-moneyness trajectory of three options over their lifetime (including the Covid crash) as well as Figure 10 which graphs the development of the size of the input data (i.e. the number of listed options) over the training period.
> Both plots are meant to illustrate the significance of the discretization-invariance of neural operators in the applicational context of volatility smoothing.
>
> ### Clarity of Experimental Figures
>
> You mention:
> > Figures 3 and 4 have unclear legends and insu!cient annotations, which reduce their interpretability. Improving their clarity and labeling would make the results more accessible and impactful.
>
> Thank you for bringing this to our attention.
> We made adjustments to Figures 3 and 4: We improved the legends and reworked their captions, hoping to have eased their interpretation.
>
> ### Analysis of Computation Complexity & Synthetic Benchmark Comparisons
>
> We thank you for the detailed comments regarding complexity and comparisons with other models:
> > The paper benchmarks its approach against SVI [1] and Ackerer et al. [2] but does not include comparisons with other key methods, such as SSVI [3] and VAE-based approaches [4]. Incorporating these would provide a more comprehensive evaluation. Additionally, using synthetic data, as in [2], could further strengthen the experimental validation.
> > While the paper highlights the elimination of instance-by-instance recalibration, it lacks a detailed discussion on computational complexity, including runtime and memory requirements. These metrics are crucial for assessing the method’s practicality in high-frequency financial contexts
>
> Regarding these considerations, we would first like to point out that SSVI is an extension of the SVI parameterisation, by making the SVI parameters functions of time-to-expiry.
> In other words, SSVI is a smooth arbitrage-free interpolation scheme for SVI slices that produces an entire surface with fewer parameters.
> In this sense, SSVI is in fact a subset of SVI, and any SSVI-benchmark is beaten by the SVI-benchmark, which explains why we focused solely on the more general SVI class.
>
> You rightly point out that our paper lacks the VAE-method as a benchmark.
> While [4] is analogous to our method, it is not applicable for index options data, which includes our S&P 500 data:
> "Standard [VAE] methods are limited to certain option markets (e.g. FX markets where options spread out on fixed rectilinear grids)" (Section 1, page 2).
> It is a unique feature of our method to be able to handle the very high-dimensional and variable-size nature of the S&P 500 input data.
> [4] uses a fixed input-size of 40, while the input size of our data increased from around 700 in 2012 to over 3000 in 2021 (this development is plotted in newly introduced Figure 10).
> We argue that this qualitative comparison is the most impactful and compelling argument in favor of our approach, while resource constraints at this stage forbid us the pursuit of retraining the GNO on synthetic data, which would have made a direct quantitative comparison possible.
>
> Finally, we agree that the implementation of a highly optimized variant (our code has prioritized prototyping and feasibility) from which to draw robust statements about HFT feasibility would be an important next step.

---

> > ### Comment · Reviewer_GefV · 2024-11-29
> >
> > Thank the authors for their response. I strongly recommend that the authors revise the aforementioned issues, particularly by supplementing relevant experimental results, which would significantly enhance the quality of this work. Although my concerns persist, I maintain that this work makes meaningful contributions to the field and, therefore, will retain my current score.

---

> ### Author Response · Authors · 2024-12-01
> **Comment [1/2]**
>
> Dear Reviewer GefV,
>
> we thank you very much for the engagement, and appreciate your confident assessment of our work and the clear recommendations for improvement.
> We understand that we haven't entirely addressed your concerns regarding computation times and benchmarking.
>
> ### Analysis of Computational Efficiency
>
> Our analysis of the computational complexity is as follows:
> - Let $\pi_\text{in}$ be the $(\tau, k)$-positions of the volatility inputs and $\pi_\text{out}$ the points at which the smoothed surface is to be computed. As part of the forward pass, the computation of the kernel integrals in each layer involves for each $y \in \pi_{\text{in}} \cup \pi_{\text{out}}$ the summation of $|\mathcal{N} (y)|  \leq K$ many terms. In total, this is the summation of  $O((n_\text{in} + n_{\text{out}}) K)$ many vectors, where $n_\text{in} = |\pi_{\text{in}}|$ is the number of input datapoints and $n_{\text{out}} = |\pi_{\text{out}}|$ is the number of queried output points.
> - The production of each of these vectors involves the evaluation of the kernel FNNs as well as its matrix multiplication with the previous hidden state $\tilde{h}_j(x)$ on the left (in our case a $16 \times 16$ matrix-vector multiplication. We take these as base units of computation and do not decompose its complexity further in terms of e.g. the hidden channel size.
> - Moreover, each layer has its own lifting network (the last one has a projection as well) and a local linear transformation (the first one hasn't), which involve $O(n_\text{in} + n_\text{out})$ many FNN evaluations/matrix multiplications.
>
> The computational effort therefore is $O(J(n_\text{in} + n_\text{out})K)$ (where $J$ is the number of layers), counted in evaluations of (relatively small) feedforward neural networks.
> We derive the following strategies to control the "time-to-smoothed-surface":
> - Changing the subsample size $K$: We are happy to be able to refer to Appendix C.6, which contains a newly introduced ablation study for $K$, and shows that there is potentially quite a bit of leeway to reduce $K$ below our level of $K = 50$. We want to thank reviewer Xa1Y at this point, who has prompted the inclusion of the ablation study into the paper.
> - Decreasing the resolution of the output surface: Lower-resolution output can be traded for faster execution speed.
> - Targeting the constants: It could be motivated to investigate smaller FNN components for kernels, liftings, and projections with faster execution speeds.
> - Accelerate compute: Since our method is purely compute-based, the use of more and/or faster GPUs will allow to process surfaces more quickly. This is a great advantage over instance-by-instance smoothing, whose fitting routines are sequential in nature and hard to accelerate.
>
> We benchmarked the execution with $n_\text{in} = 897$ and $n_\text{out} = 2500$ (50x50 output grid) on a consumer-grade laptop CPU as well as an Nvidia Quadro RTX 6000 GPU:
>
> | $K$ | 10      | 20      | 30      | 40      | 50      |
> | --- | ------- | ------- | ------- | ------- | ------- |
> | CPU | 74.6 ms | 149 ms  | 215 ms  | 273 ms  | 336 ms  |
> | GPU | 3.57 ms | 6.93 ms | 10.0 ms | 13.2 ms | 16.7 ms |
>
> These numbers confirm the linear scaling of compute in $K$ and the great accelerability of our method using GPUs.
> And, they show that computation times for typical grid sizes are fast enough for performing smoothing several times per second, which is more than sufficient for most applications (we elaborate on this further below).
>
> [Continued in the next comment]

---

> ### Author Response · Authors · 2024-12-01
> **Comment [2/2]**
>
> [Continuing previous comment]
>
> Following your advice we will include and expand on these considerations in the next revision of our paper (ICLR timelines do not allow to amend the PDF right now).
> We want to contextualize the breadth of these considerations as follows:
> - We have been approaching the implications of our work from a different angle: We argue that -- at this stage -- the most significant broad impact of our approach for volatility smoothing is that it "lowers the entry-barrier for effective volatility smoothing" (page 2), by moving to the operator level and thereby unlocking the learnings from the entire historical dataset in the smoothing of every single surface. The trained operator could then be "served as a hands-off tool, could provide cheap and accurate surfaces for use in downstream tasks". Most practitioners perform implied volatility smoothing in intervals of a few minutes, if not only a few times during every day, and we believe that our data-focused method, in its current state, could already be highly transformative for these individuals. All performance concerns in this context are dispelled by taking the scalability of compute for granted. In fact, by moving the computations from the CPU to a single GPU, we accelerate our method by a factor of 20 (compare above table).
> - As indicated by the experiments, our implementation allows for smoothing at a frequency of seconds. This may be sufficient for many applications. On the other hand, we are not in the position to draw robust conclusions on the feasibility for the full range of HFT applications from our current implementation and its complexity analysis. Our implementation has not been prioritizing speed but feasibility (in particular the graph construction is brute-forced and takes around 100 ms). While our method delivers smoothed surfaces on GPUs within a few milliseconds, HFT nowadays operates at sub-millisecond timescales. Scope and resource constraints for this paper forbid us the pursuit of a highly optimized architecture for fast paced algorithms, and find that such would be appropriate for future work. Moreover, such specialization would probably come at the cost of the generalization capabilities of the method, limiting the "foundational-ness" of our method, which we have been prioritizing. However, we would like to emphasize again that for the most common practical applications, smoothing results within a few milliseconds are sufficient.
>
> ### Experiments using synthetic data
>
> The independence of our method with respect to the input grid make the following experiment possible:
> - We take SSVI as formulated in [5].
> - We compute implied volatility for some SSVI parameters on two synthetic rectilinear grids for  $(\rho_\text{min}, \rho_\text{max}) \times (z_\text{min}, z_\text{max})$: One of size $10 \times 50$ and another of size $100 \times 100$
> - We feed the sparse size-$10 \times 50$ data into the graph neural operator, which smooths this data to align with the dense, uniform $100 \times 100$-grid.
> - We compare the error between the GNO output and the SSVI input on the size-$100 \times 100$ grid
>
> For the average historical SSVI parameters for the S&P500 index as reported in Table 4 of [5], we achieve a mean absolute percentage error of $\delta_\text{abs} = 2.03\%$.
> We believe that it is a great display of the generalization capabilities of our method, that it is able to smooth synthetic SSVI data on a synthetic grid with an average relative error of around 2%, without ever having been trained on this data.
>
> We will include an exhaustive description of this experiment into the final revision of our paper.
>
>
>
> Finally, we want to thank you again for reiterating on your concerns.
> They have prompted above discussion, whose inclusion into the paper we believe significantly strengthens and rounds off its presentation.
>
>
>
> [5] Lind, Peter Pommergård, and Jim Gatheral. “NN De-Americanization: A Fast and Efficient Calibration Method for American-Style Options.”

---

### Meta-Review · Area_Chair_rVqL · 2024-12-20

**Metareview:**

The authors proposed a neural operator approach for implied volatility surface smoothing, and the trained model can directly map observed data to implied volatility surfaces, eliminating the need for calibration every time. Some reviewers pointed out missing baseline methods, such as the VAE approach, which was a flaw, but not very crucial. I recommended an acceptance as all reviewers left positive feedback in the end.

**Additional Comments On Reviewer Discussion:**

One reviewer raised the score after rebuttal.

---

### Decision · Program_Chairs · 2025-01-22

Accept (Poster)